# Humans display interindividual differences in the latent mechanisms underlying fear generalization behaviour

Kenny Yu [1✉], Francis Tuerlinckx [1], Wolf Vanpaemel [1] & Jonas Zaman [1,2]

Human generalization research aims to understand the processes underlying the transfer of prior experiences to new contexts. Generalization research predominantly relies on descriptive statistics, assumes a single generalization mechanism, interprets generalization from mono-source data, and disregards individual differences. Unfortunately, such an approach fails to disentangle various mechanisms underlying generalization behaviour and can readily result in biased conclusions regarding generalization tendencies. Therefore, we combined a computational model with multi-source data to mechanistically investigate human generalization behaviour. By simultaneously modelling learning, perceptual and generalization data at the individual level, we revealed meaningful variations in how different mechanisms contribute to generalization behaviour. The current research suggests the need for revising the theoretical and analytic foundations in the field to shift the attention away from forecasting group-level generalization behaviour and toward understanding how such phenomena emerge at the individual level. This raises the question for future research whether a mechanism-specific differential diagnosis may be beneficial for generalization-related psychiatric disorders.

[1] KU Leuven, Leuven, Belgium. [2] University of Hasselt, Hasselt, Belgium. ✉email: kenny.yu@kuleuven.be

As humans, we frequently encounter instances of 'once bitten, twice shy' in our daily lives. Having been bitten by a dog, we may naturally be wary of other dogs in the future. This capability of generalizing prior learning to unfamiliar situations is crucial because it enhances our adaptivity in a constantly changing environment.

The fear of a particular dog, however, might become rigid and harmful if it is spread to a wide range of harmless dogs or dog-like animals. Maladaptive generalization behaviour is associated with several psychopathologies, such as anxiety disorder[1–3], autism[4], and obsessive-compulsive disorder[5,6]. Consequently, assessing generalization behaviour and quantifying the proclivity of a latent generalization process is an important goal of generalization research.

Typically, generalization is studied using a conditioning paradigm[7–9]. In a generalization experiment, participants first learn about the relationship between one or more cues (conditioned stimulus, CS; e.g., a circle) and their associated consequences (unconditioned stimulus, US; e.g., an electrical shock). Following that, generalized responses are obtained by measuring conditioned responding to multiple novel test stimuli (TS) that most often differ slightly in certain physical dimensions from the stimuli used during training (e.g., the size or color of a circle). As the physical resemblance to the CS diminishes, the response strengths tend to decrease[7,9,10]. This relationship is presumed to reflect a latent similarity-based generalization process, with a flatter response gradient implying a greater proclivity for generalization. Traditionally, researchers in the field have focused on the group gradient, which is the average of the individual response gradients[7]. Along with the summary statistics, numerous statistical models and theories were applied to quantitatively describe and predict generalization behaviour[10–16].

The currently dominant approach of inferring a generalization mechanism (i.e., the propensity to generalize past learning to encountered unfamiliar stimuli) directly from the observed (averaged) response gradient has several limitations. One is that it relies on the implicit presumption of a single mechanism (i.e., generalization) that underlies the behaviour and thus does not consider the possibility that different latent mechanisms can yield the same observable behaviour.

However, such a single mechanism perspective is not realistic. Generalization can be regarded as a cognitive process whereby previous learning is transferred to newly encountered stimuli based on their similarity to the originally learned stimulus[11,12]. Thus, if any aspect of the process - from the initial learning to stimulus perception to the actual transfer of learning - goes awry, it may result in aberrant generalization behaviour. For instance, it has been observed that manipulating the experimental learning experiences, such as the number of learning trials, the reinforcement rate, and the learning procedures, exerts a direct influence on generalization gradients[17–21]. Without proper learning, organisms are incapable of generating distinct and consistent responses. However, even when CS-US learning is firmly established, generalized responses can still occur as a result of imperfect perceptual discrimination between the CS and newly encountered stimuli[22–26], providing the second potential mechanism for generating flatter or noisy response gradients. In earlier research, the relationship between insufficient perceptual discrimination and generalization behaviour has been discussed[27,28]. Recently, researchers have further pointed out the relationship between stochastic perception and generalization behaviour. The implication is simple: a generalized response to TSs may not necessarily emerge due to a generalization process but merely emerge due to an inability to perceive the TS as different from the CS. In support of this idea, the high prevalence of problematic stimulus discrimination during a generalization

protocol was demonstrated with strong effects on the strength of generalized responding. Conditioned responses to TSs were strong when they were falsely perceived as the CS while attenuated when the stimulus was perceived as different from the CS (even during CS trials)[22–24,26]. This inflated the inferred extent of a latent generalization process as response gradients were much narrower after accounting for perceptual errors[25]. Yet, the traditional approach predominantly attributes differences in generalized responding to a single generalization mechanism, or in some instances, neglects to acknowledge any underlying mechanisms altogether. As a result, the inherent variability of the aforementioned mechanisms and their contribution to the observed behaviour is often disregarded[29]. Additionally, it is pertinent to note that certain patient populations often exhibit impaired learning[30–33], misperceptions[34–36], and flatter generalization[1–6] patterns when compared to healthy controls. Even at the pre-clinical level, variations in these underlying processes have been associated with state anxiety levels[37,38].

Under the assumption that generalization behaviour is not restricted to a single cognitive mechanism accounting for all observed behavioural variation, all relevant psychological tasks (e.g., learning, perception, generalization) should be studied concurrently. Neglecting the influence of any of these aspects renders a comprehensive understanding of generalization behaviour impossible. Inspiration can be found in the category learning literature[39], in which it is assumed that the performance of categorization can be influenced by similarity judgment. With similarly-rating data, the mental representations of physical differences between stimuli are derived. These representations will ultimately affect how organisms make category decisions, as observed in the category learning data. Without the use of multi-source data in generalization research, insight into the potential (multiple) mechanisms at play will remain limited as the same observable behaviour may stem from various latent processes with no way to differentiate and identify among them.

Another notable limitation of previous fear generalization research is the disregard of individual differences, while they actually may be a rich source of information about the hidden causes that drive individual generalization gradients. In recent research, it has been observed that generalization behaviour has diverse shapes across and within individuals[23,25,26]. In many instances, aggregated group data will not adequately reflect individual behaviour, potentially leading to biased conclusions about behaviour phenomena[40]. An abundance of empirical evidence has shown that individual differences in learning and perception may account for the diversity in generalization behaviour[29]. As an example, a recent study has provided evidence that the use of inter-individual differences in perception and memory as predictors substantially improved the ability to account for differences in generalization gradients[41]. In the field of perception, it is well-acknowledged that humans do not have direct access to the physical world, and that the brain may probabilistically represent sensory inputs with large individual differences in how one perceives their surroundings[42–48]. However, previous research on generalization has largely ignored the potential impact of inter-individual differences, often by either equating perception to the physical dimension (making perception the same for everyone) or employing psychophysical mapping methods such as multidimensional scaling[11,12] to derive a mental representation of encountered stimuli. These psychophysical mapping methods mostly assume a universal (and invariant) psychological space, with stimulus representations in an identical constellation for all humans, rendering them unable to account for inter- and intra-individual variations in perception, the impact of learning on perception[49–53] and the impact of both on generalized responses[54–56]. Alike, despite the robust empirical

evidence that individuals do not achieve the same level of learning at the end of most experiments[17–21], these differences are surprisingly often overlooked when studying subsequent differences in generalization behaviours. Such oversight is not without pitfalls, as it may lead to incorrect inferences regarding the extent of underlying generalization mechanism.

In this paper, we go beyond these limitations by proposing and implementing a computational model in which differences in generalization behaviours can arise as a result of differences in learning, perception, and a generalization propensity or by a combination of them. A flat gradient, previously thought to reflect solely a propensity to overgeneralize can here stem from a lack of learning, perceptual errors, and/or a large generalization propensity. The computational model is, in essence, a formalized theory of generalization that takes different behaviour-generating mechanisms and their interactions into account. The basic tenet of the model is that generalization behaviour should be considered as a dynamic (non-static) system involving numerous variables (e.g., stimulus perception, US expectancy in learning and generalization stages) and parameters (e.g., learning rate and generalization rate), which are mathematically and computationally interconnected. We postulate that the conditioned response to the CS is time-dependent and will be adjusted in accordance with the learning process and that the extent to which generalization occurs is governed by the individual's inclination for similarity-based generalization based on the perceptual or physical resemblance of the encountered stimulus to the CS. The use of time-resolved perceptual data opens the possibility to investigate the influence of perceptual variability on generalization. With this model, an integrated analysis of data from various cognitive and perceptual tasks relevant to generalization behaviour is possible.

Before elaborating the computational model further, we first discuss the data that are used to test our model. We used two fear generalization data sets (Experiment 1: $N = 40$, Experiment 2 : $N = 40$). Both experiments employed circles of varying sizes as CS and TS, with a painful electric shock serving as US (for a detailed account, see the *Method* section). During each trial in both the learning and generalization phases of each experiment, participants had to estimate the stimulus size (i.e., perceptual data) and provide US expectancy ratings (i.e., learning and generalization data, see Fig. 1). Experiment 1 (mean age = 21.8 years, SD = 5.3, 26 females (65%)) used a simple fear conditioning (i.e., one cue preceding a painful US) procedure with a lower reinforcement rate (50%), while in Experiment 2 (mean age = 23.5 years, SD = 8.9, 26 females (60%)), differential fear conditioning was adopted (i.e., two cues, one preceding a painful US, one predictive of the absence of pain) and a higher reinforcement rate (83%). With Experiment 2 compared to 1 differing on aspects that should foster learning, we can examine if the model can capture different patterns of generalized behaviour of participants under very different learning experiences.

Our computational model is implemented as a Bayesian multilevel mixture model on the collected multi-source data (i.e., learning, perception, and generalization) and contains four important properties. First, given the complexity of generalization that may emerge from multiple mechanisms, we employ Bayesian statistics to characterize our uncertainty about the parameters as probability distributions in a principled manner[57,58]. This allowed us to assess the effect of different psychological processes on human generalization, taking into account the available evidence. Second, the multilevel structure is implemented to account for the ubiquitous quantitative individual differences by inferring parameter values from both the individual and group levels[59–62]. Third, in the model, generalization behaviour is not restricted to a single generating process but could emerge from several different latent processes situated at different levels (e.g., learning, generalization). Fourth, we incorporate a mixture framework[59,63,64] that allows us to allocate individuals into potential clinical-relevant subgroups based on the malfunctioning of certain latent processes. The specific pattern of generalized behaviour can be the result of (1) problematic learning (i.e., Non-Learners), and when learning occurred, (2) an extreme generalization tendency (i.e., Overgeneralizers), or (3) a similarity-based generalization process where perceptual variability does not impact stimulus similarity (i.e., Physical Generalizers), or (4) a similarity-based generalization process where differences in stimulus perception influence the extent of stimulus similarity (i.e., Perceptual Generalizers). This tree-like structure naturally leads to four groups.

Figure 2 displays the general model structure. Associative learning and similarity-based generalization are the two presumptive processes that lead to final generalization behaviour. The two processes encapsulate how participants acquire the CS-US associations and consequently generalize learning to newly encountered stimuli according to their level of similarity based on either their physical or perceptual distance. The two most important parameters in the model that embedded psychological processes are the learning rate $\alpha_i$ and generalization rate $\lambda_i$. The former represents how individuals constantly update their expectations about the CS-US association, with higher values indicating more learning occurs with every CS interaction. The latter represents how individuals generalize their learning to novel stimuli according to the distance between the CS and the novel stimulus, with lower values indicating more generalization occurs for a fixed stimuli distance. However, the degree to which these two processes play a role in generalization behaviour can vary among individuals. The model distinguishes between four groups. The group membership parameter $m_i$ ($m_i = 1, 2, 3, 4$ corresponds to Non-Learners, Overgeneralizers, Physical Generalizers, and Perceptual Generalizers, respectively) indicates the latent group of participants according to their behaviour patterns in learning, perception, and generalization.

The proposed model not only examines the impact of distinct latent processes on generalized responses but also identifies potential clinical subgroups among participants by accounting for specificities in their latent processes. Furthermore, it parameterizes the latent similarity-based generalization process (in straightforward index for generalization propensity) in such a manner that it enables the comparison between individuals despite their differences in learning and perceptual sensitivity. Focusing on the behaviour-generating process, the model investigates how different presumptive mechanisms combined (i.e. learning, perception, generalization) influence the generalization behaviour patterns of different individuals.

## Methods

To disentangle the influence of learning, perception, and generalization propensity on generalization behaviour, we used the data of two pre-registered experiments with similar designs but different conditioning paradigms. Experiment 1 used a simple conditioning paradigm in which participants learned the association between a single cue (CS+) and the occurrence of an electric shock (US) with a 50% reinforcement rate. In Experiment 2, participants learned not only the association between the CS+ and US, but also the disassociation between another cue (CS-) and US (i.e., differential conditioning). We expected more learning to occur in Experiment 2 due to a higher reinforcement rate (83%)[65] and the additional safety cue[66].

The two data sets used in the experimental study are taken from two studies that were pre-registered on the Open Science Framework at 2017 (OSF; Experiment 1: https://osf.io/b4ngs;

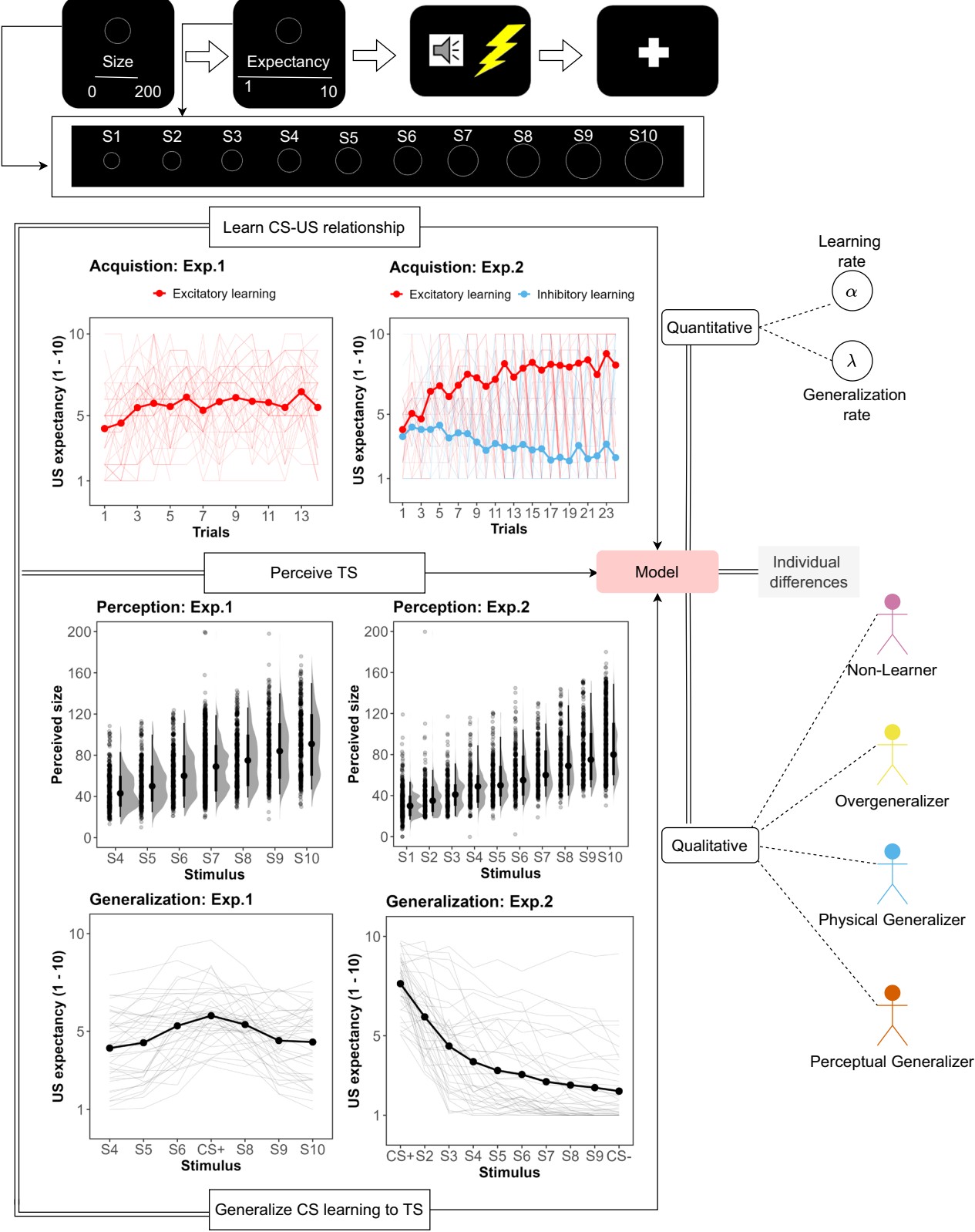

**Fig. 1 Experimental design and data.** Overview of the experimental paradigms, stimulus, and size estimation, acquisition, and generalization data of two experiments.

Experiment 2: https://osf.io/t4bzs). These studies focused on different hypotheses and research questions and used different methodologies than those of the current study, so the plans for descriptive statistical analyses included in the protocol are not

relevant for the current study. As the current research questions and corresponding analysis method, which is based on computational modelling, were not included in this, or any other, pre-registration protocol, the current study should be considered as

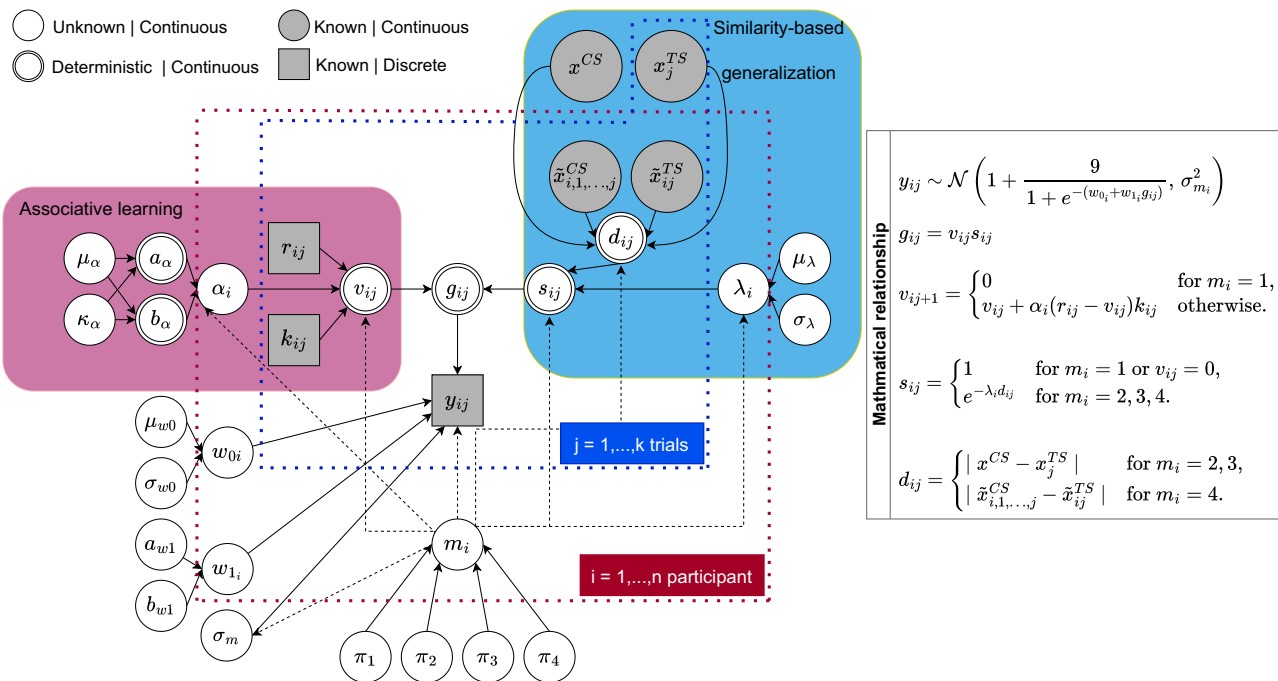

**Fig. 2 The Directed Acyclic Graph representing the Bayesian computational model.** The relationship among different variables and parameters in the model is displayed. The data for modelling includes US expectancy ratings, denoted as $y_{ij}$. These ratings follow a normal distribution with a mean of $\theta_{ij}$ and a variance of $\sigma^2$. The mean US expectancy, $\theta_{ij}$, is obtained by applying a sigmoid function to the generalized associative strength $g_{ij}$. The generalized associative strength encompasses learning and similarity-based generalization processes.

being non-preregistered, as far as hypotheses and analyses are concerned. The raw data for the two experiments, codes for the computational model and analysis, and supplementary information, which contain more modelling details, are available at another OSF repository: https://osf.io/sxjak/.

**Experiment.** To reach the pre-registered sample size of 40 complete data sets in both experiments we had to recruit 43 and 40 participants, respectively, as three participants in the first experiment did not complete the task due to technical problems (Experiment 1: mean age = 21.8 years, SD = 5.3, 26 females (65%); Experiment 2: mean age = 23.5 years, SD = 8.9, 26 females (60%)). The final data set used in the analysis for both experiments comprised 40 participants. All participants were recruited via the KU Leuven's Sona Systems and received either course credits or a monetary compensation (€12 for Experiment 1 and €16 for Experiment 2). Informed consent was provided by participants at the beginning of both experiments. Participants were instructed in English. They were asked to report their gender. The study was approved by the KU Leuven's Social and Societal Ethics Committee (G-201610641).

In both experiments, the conditioned stimuli (CSs) and test stimuli (TSs) were circles varying in size with white outlines against a black background. The size dimension of circles has been widely employed to test the generalization of a trained fear response in both healthy volunteers and clinical populations[2,22,23,26,67–69]. The stimulus set as a whole consisted of ten circles (S1–S10) ranging in diameter from 50.80 to 119.42 mm, with steps of 7.624 mm in between. Experiment 1 used seven circles (S4–S10), with the middle one (S7; 96.54 mm) serving as the CS+ and the remaining six stimuli serving as TSs. Experiment 2, with a differential learning paradigm, employed the whole stimulus set (S1–S10). The CS+ and the CS- were counterbalanced between individuals, with the smallest (S1; 50.8 mm) and biggest (S10; 119.42 mm) circles serving as either the CS+ or

the CS-. The remaining nine stimuli served as TSs and were only presented during the generalization phase.

Both experiments involved a noxious electrocutaneous stimulus as the unconditioned stimulus (US). Electrocutaneous stimuli were delivered through an electrocutaneous stimulation device (Constant Current Stimulator, model DS7) through two Ag/AgCL electrodes (8mm), filled with K-Y gel on the wrist of the non-dominant hand. In the calibration phase, the US intensity was adjusted to pain tolerance levels for each participant using the Ascending Method of Limits approach[70]. Participants were instructed that the stimulation should be painful but tolerable. The targeted rating for each stimulus was 8 on a visual analog scale (VAS; 0 = no pain, 10 = worst imagined pain). For Experiments 1 and 2, the average intensity of the electrical stimuli was 19.4 mA (SD = 9.49) and 22.7 mA (SD = 7.72), respectively.

Both experiments comprised four phases: calibration, practice, acquisition, and generalization. Before the experiment started, participants were given oral and written instructions and informed consent was obtained. After calibrating the intensity of the US, participants completed six practice trials to habituate to the task. In the practice trials, the CS(s) were displayed on the computer screen for 10 seconds while a size Visual Analogue scale (VAS, labels: 0–200 millimeters) was displayed at the bottom for participants to indicate the perceived size of the presented stimulus. Participants were not given feedback on their size estimations. Exactly 5 seconds after the onset of the size-VAS, it was replaced by an expectancy-VAS (labels: no shock = 1, definitely a shock = 10) on which participants rated their expectation of being shocked given the presented stimulus. After an additional 5 s, the stimuli and the expectancy-VAS disappeared and a fixation cross appeared for 7–10 s (intertrial interval, ITI). Acoustic startle probes (105dB noise for 0.05s) were here presented on each trial either within 6–9 s after trial onset or within 4–7 s after ITI onset to habituate to the startle probe.

Following the practice phase, the acquisition phase started, with 14 CS+ trials for Experiment 1 and 24 trials (12 CS+ trials

**Table 1 Priors specification**

| Parameter | Prior | Hyperprior |
|---|---|---|
| Learning rate | $\alpha_i = 0$, for $m_i = 1$ <br> $\alpha_i \sim \text{Beta}(a_\alpha, b_\alpha)$, otherwise | $a_\alpha = \mu_\alpha \kappa_\alpha$ <br> $b_\alpha = (1 - \mu_\alpha) \kappa_\alpha$ <br> $\mu_\alpha \sim \text{Beta}(1, 1)$ <br> $\kappa_\alpha \sim \text{Uniform}(1, 10)$ |
| Generalization rate | $\lambda_i = 0$, for $m_i = 1$ <br> $\lambda_i \sim N(\mu_\lambda, \sigma_\lambda^2) T(0, .0052)$, for $m_i = 2$ <br> $\lambda_i \sim N(\mu_\lambda, \sigma_\lambda^2) T(.0052, \infty)$, otherwise | $\lambda_\mu \sim N(.1, 1) T(0, \infty)$ <br> $\lambda_\sigma \sim \text{Uniform}(10^{-9}, 1)$ |
| Baseline response | $w_{0i} \sim N(\mu_{w_0}, \sigma_{w_0}^2)$ | $\mu_{w_0} \sim N(0, 10^2)$ <br> $\sigma_{w_0} \sim \text{Half-Cauchy}(0, 2)$ |
| Scaling | $w_{1i} \sim \text{Gamma}(a_{w_1}, b_{w_1})$ | $a_{w_1} \sim \text{Half-Cauchy}(0, 2)$ <br> $b_{w_1} \sim \text{Half-Cauchy}(0, 2)$ |
| Response noise |  | $\sigma_1 \sim \text{Uniform}(10^{-9}, 1.5)$, for $m_i = 2, 3, 4$ <br> $\sigma_2 \sim \text{Uniform}(1.5, 3)$, otherwise |
| Latent group allocation | $m_i \sim \text{Multinomial}(1, \pi_1, \pi_2, \pi_3, \pi_4)$ |  |
| Group probability |  | $\pi_1, \pi_2, \pi_3, \pi_4 \sim \text{Dirichlet}(c(1, 1, 1, 1))$ with $\sum_{i=1}^{4} \pi_i = 1$ |

The notation T(s,t) denotes truncation that limits the probability distribution to the range between s and t.

and 12 CS- trials) for Experiment 2. Trial structure was identical to the previous phase, apart from US administrations at CS(s) offset in either 50% (Experiment 1) or 83% (Experiment 2) of the CS+ trials and the frequency of startle probes (only in 44% and 48% of the trials in Experiment 1 and 2). CS- trials were never paired with the US. The generalization phase in Experiment 1 and 2 comprised 4 and 3 blocks, respectively, separated by a 3-minute break. The US was never paired with the CS- or the TS trials. Each block in Experiment 1 comprised 22 CS+ trials and 24 TS trials and always started with 10 consecutive CS+ trials (i.e., reacquisition) to prevent extinction of the conditioned response. Experiment 2 had 14 CS+ trials, 8 CS- trials, and 32 TS trials in each block, starting with 6 consecutive CS+ trials. In both experiments, the initial block of the generalization phase did not include any reacquisition trials, as it immediately followed the acquisition trials. Consequently, the generalization phase of the two experiments encompassed a total of 174 (i.e., (22 + 24) 4 − 10) and 156 (i.e., (14 + 8 + 32)3 − 6) trials, respectively.

Startle eyeblink responses, US expectancy ratings, and size estimations were collected during acquisition and generalization phases in both experiments. However, the learning component in the model has a time-dependent structure that considers information from all the previous trials, and we had sparse assessment for the startle data, resulting in varying quantities of available responses across individuals. Therefore, startle data were not evaluated and only US expectancy data were used to represent fear responses to avoid drawing biased conclusions for inter-individual differences.

**Model.** In this paper, we use a Bayesian multilevel mixture model that incorporates several psychological mechanisms underlying human generalization behaviour. Specifically, the goal of the model is to explain the observed US expectancy ratings during acquisition and generalization phases by a combination of learning and generalization processes. The statistical inference is done using Bayesian statistics. Figure 2 provides a graphical representation of our model using a Directed Acyclic Graph[71,72]. The model contains several components, which will be discussed in detail in the following sections. In this section, we will explain the assumptions and the general structure of the model.

The relevant data for modelling are the US expectancy ratings, denoted as $y_{ij}$ (where $i = 1, \ldots, n$ refers to the participant and $j = 1, \ldots, k$ to the trial). As shown in Fig. 2, they are assumed to be normally distributed with mean $\theta_{ij}$ and variance $\sigma^2$. The mean US expectancy $\theta_{ij}$ is a nonlinear transformation (by the sigmoid

function) of the generalized associative strength $g_{ij}$, that represents two core processes: learning and (similarity-based) generalization.

Our model uses stimulus size estimation (perception) and US expectancy data collected from both the acquisition and generalization stages in the experiment and specifies the data-generating process by which participants first learn to expect the US after the CS(s) and then generalize it to the TS(s), based on stimuli similarity. This similarity can either be physical or perceptual, thus making room for perception in generating generalization behaviour. There are several sets of qualitatively different plausible assumptions regarding how learning and similarity-dimension influence generalization, and they may lead to different models. As a consequence, we need to use model selection to decide which model is the best representation for the underlying generalization process of an individual. In this paper, model selection is considered as a parameter estimation problem for the latent group membership parameter, by creating a mixture model that has several alternative models as sub-models. The parameter estimation procedure will allocate individuals to the four latent groups (in a probabilistic fashion) and thus create a posterior distribution of the group membership, informing for every individual which model is supported by the data the most. The mixture framework in our model enables the identification of sub-response patterns within the entire population. In Fig. 2 and Table 1, some equations and prior distributions are conditioned on the discrete latent variable $m_i$ ($m_i = 1, 2, 3, 4$), where $m_i$ indicates the group membership for individual $i$. The four latent groups are: Non-Learners, Overgeneralizers, Physical Generalizers, and Perceptual Generalizers (corresponding to the labels 1, 2, 3, and 4, respectively). There are different mechanisms at play in each group that ultimately lead to the final response, implying multiple sources for generalized responding.

As depicted in Fig. 2, the model consists of several parameters. Some are general (i.e., they apply to all individuals), while others are person-specific and intended to represent quantitative individual differences. In this study, the parameters $\alpha_i$ and $\lambda_i$ characterize the two latent processes that influence the generalization behaviour of participants in the two experiments: learning and generalization, respectively. In a Bayesian framework, prior distributions are assigned to all model's parameters and they represent the relevant information about these parameters prior to the observation of the data. Table 1 contains the prior specification for both general and person-

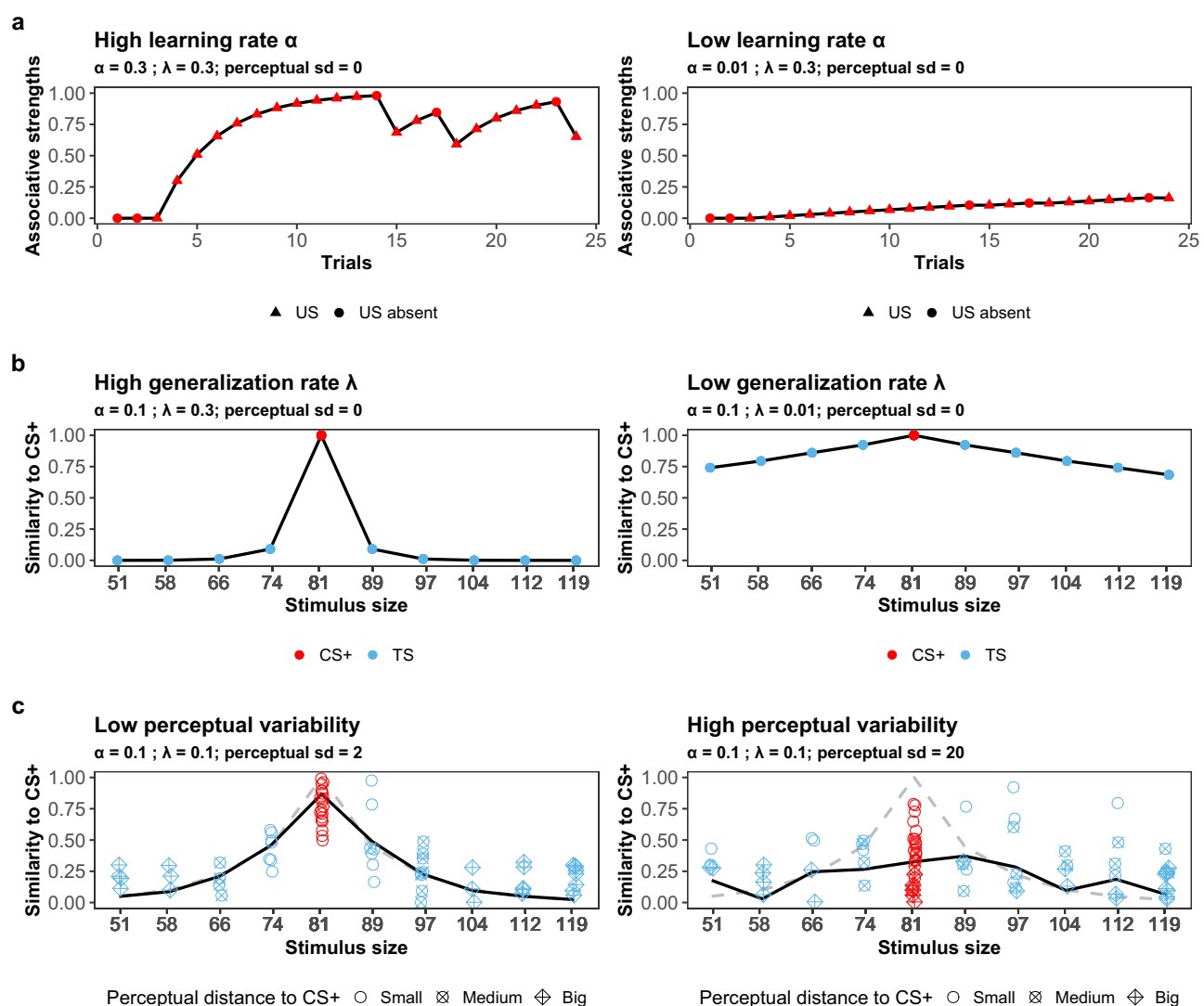

**Fig. 3 Simulation of different latent mechanisms.** This figure illustrates the role of the several mechanisms in the computational model that contribute to the observed generalization behaviour. In panel **a**, it is shown how the learning rate parameter influence the learning patterns. The higher the learning rate $\alpha_i$, the more learning happens after the CS trials. In panel **b**, the influence of the generalization rate parameter $\lambda_i$ is depicted. The lower the generalization rate $\lambda_i$, the more generalization happens for the same stimulus (i.e., given a constant stimuli distance). In panel **c**, the influence of perceptual variation is illustrated. More perceptual variations will result in more erroneous similarity judgments between the CS+ and TSs. The solid black curve represents the generalization gradient with perceptual variations, and the dashed gray curve represents the generalization gradient with perfect perception. Noted that perceptual sd is not a parameter in the model, we assessed perceptual variability from the size estimation task in the experiments.

specific priors. The multilevel structure in this study implies that person-specific parameters are assumed to come from an overarching population distribution. Such a population distribution can be considered as a prior distribution with unknown parameters. For these parameters governing the population distribution, we assign weakly informative hyperpriors (representing a lack of knowledge). After observing the data, the (hyper)priors are updated to the posterior distributions of parameter values, reflecting the adjusting of our prior beliefs in light of the newly acquired information. In what follows, we will explain the several major components of the model (indicated by a background color in Fig. 2).

The first major component of the model is associative learning, which reflects the changing associative strengths of the CS-US relationship at different time points. In our model, we assume that the establishment of associative strength is determined by the error-driven Rescorla-Wagner learning rule[73]. According to the rule, the associative strength of the CS at the

current time point, which captures the expectation that it will be followed by a certain outcome (US), is determined by a learning rate parameter ($\alpha$) and the discrepancy between what was expected and experienced at the previous CS presentation (i.e., prediction error). The larger the prediction error, the larger the adjustment of one's expectation; the larger the learning rate, the greater the amount of learning for a given prediction error (see Fig. 3, panel a). In Experiment 2, participants learned to predict the US when encountering the CS+ (i.e., excitatory learning) and to predict US absence when encountering the CS- (i.e., inhibitory learning), whereas Experiment 1 only involves excitatory learning. Mathematically, the applied learning rule in our model postulates that:

$$v_{i,j+1} = \begin{cases} 0 & \text{for } m_i = 1 \,(\text{i.e., the nonlearners}), \\ v_{ij} + \alpha_i(r_{ij} - v_{ij})k_{ij} & \text{otherwise (i.e., the 3 other groups).} \end{cases}$$

(1)

where $v_{ij}$ is the associative strength of the CS in trial $j$ for participant $i$, inferring the time-dependent learned expectation of the CS(s) - US association ($v_{ij}^+$ for the CS+ and $v_{ij}^-$ for the CS- in the differential learning experiment); $k_{ij}$ is a dummy variable to control the occurrence of updating, $k_{ij} \in \{0, 1\}$, where 1 indicates updating to happen and 0 otherwise. The role of the dummy variable $k_{ij}$ is to ensure that learning only happens during the CS trials; $r_{ij}$ is a variable with trial outcomes (i.e., US or no US), $r_{ij} \in \{0, 1\}$ for the CS+ and $r_{ij} \in \{-1, 0\}$ for the CS-. The parameter $\alpha_i$ corresponds to the learning rate and regulates the amount of value learning adaptation for individual $i$ ($\alpha_i \in [0, 1]$, with higher values representing more learning from the prediction error). As implied above, it is assumed that the associative strength updating process only happens during the CS trials. When neither the CS+ nor the CS− is presented, neither the associative strengths for the CS+ nor the CS− will change.

In our model, we also allow for the possibility that no learning occurs. We do this by including the parameter $m_i$ for each individual $i$. For Non-Learners (i.e., $m_i = 1$), no associative strength is acquired.

In the model, it is assumed that the learned response to a particular stimulus generalizes to another stimulus based on the similarity between these stimuli in mental space, denoted as $s_{ij}$ for individual $i$ at trial $j$. Unlike most generalization research where the individual variability in learning is not considered, we assumed that the amount of the generalized response is time-dependent and will be adjusted in accordance with the current associative strength as defined in Eq. (1):

$$g_{ij} = v_{ij} s_{ij}. \tag{2}$$

For differential learning, in line with the notion that response gradients are congruently affected both by excitatory and inhibitory learning, the final associative strength to a certain TS is the summation of the generalized excitatory strength and the generalized inhibitory strength[74]: $v_{ij}^+ s_{ij}^+ + v_{ij}^- s_{ij}^-$.

The similarity $s_{ij}$ is assumed to decrease exponentially as the distance $d_{ij}$ between stimuli increases, in accordance with Shepard's law[11,12]:

$$s_{ij} = e^{-\lambda_i d_{ij}}. \tag{3}$$

The decay parameter $\lambda_i$ reflects the generalization propensity of the $i$ individual: Larger values of $\lambda_i$ lead to a faster decay of similarity with increasing distance (see Fig. 3, panel b). Note that a flat response will be independent of $\lambda_i$ if learning did not happen in the first place (i.e., when $v_{ij} = 0$).

The vast majority of contemporary research on human generalization analyzes generalized responses along the physical dimension of the manipulated stimulus features, completely ignoring the potential influence of mental (or perceptual) representations. In that case, $d_{ij}$ in Eq. (3) is the physical distance between the TS and the CS at trial $j$ for individual $i$, which is invariant between and within individuals.

Recent research on generalization[22–26] has demonstrated the existence of perceptual variation between and within individuals. Hence, perceptual variation may affect inter-stimulus similarity if perception, rather than physical characteristics, determines the coordinates of stimulus representations in a multidimensional psychological space. As a consequence, the similarity between two stimuli will no longer be constant, but will instead vary according to the perception of the individual. Depending on the nature of the warping of the physical dimension through perception, the mental similarity between a pair of stimuli along the continuum either decreases or increases, with corresponding effects on the extent of generalized responding (see Fig. 3, panel c).

The current model entertains two qualitatively distinct possibilities of how similarity-based generalization can occur and both are represented as components in the mixture model. First, it may be the case that generalization occurs as physical stimulus distances decrease. This is the case for individuals with $m_i = 3$, who are classified into the Physical Generalizers group. Second, it assumes that generalization occurs in conjunction with inter-stimulus distances depending on the stimulus perception of individuals. These individuals have $m_i = 4$ and are classified into the Perceptual Generalizers group.

Mathematically, the similarity-based generalization process in our model can be written as:

$$s_{ij} = \begin{cases} 1 & \text{for } m_i = 1 \text{ or } v_{ij} = 0, \\ e^{-\lambda_i d_{ij}} & \text{for } m_i = 2, 3, 4. \end{cases} \tag{4}$$

In the model, an individual $i$ whose response data demonstrate both learning and similarity-based generalization can be assigned to either the Physical Generalizers ($m_i = 3$) or Perceptual Generalizers ($m_i = 4$) groups. The learned CS-US associative strengths ($v_{ij}^+$ or $v_{ij}^-$) of these individuals will decrease as the physical or perceptual distance between the CS and a TS increases, with a ratio dictated by the generalization rate parameter $\lambda_i$ (see Fig. 3, panel b). Hence, if $m_i = 1$ (where learning is absent), the similarity is set to 1. Otherwise, the similarity is a function of the distance between the CS and the TS. When $m_i = 2$, $\lambda_i$ takes an extremely small value, ensuring that the similarity remains higher than 0.7 for all physical distances in the experiment. We set the upper boundary for $\lambda_i$ for the Overgeneralizers ($m_i = 2$) group based on the criterion that the learned response ($v_{ij}$) remains at least 70%, even when encountering the most physically distant stimulus to CS(s) (i.e., distance = 68.62). This criterion returns $\lambda_i = .0052$ and it also serves as the lower boundary for both the Physical Generalizers ($m_i = 3$) and Perceptual Generalizers ($m_i = 4$). Therefore, $\lambda_i \in [0, .0052]$ and $\lambda_i > .0052$ for Overgeneralizers and Physical Generalizers or Perceptual Generalizers, respectively (see Supplementary Figs. 1 and 2). When $m_i = 3$, the distance is defined as an absolute (invariant) difference between the coordinates of the CS and the TS:

$$d_{ij} = \left| x^{CS} - x_j^{TS} \right|, \tag{5}$$

with $x^{CS}$ the coordinate of the CS and $x_j^{TS}$ the coordinate of the TS on trial $j$, both on the physical dimension. In contrast, when $m_i = 4$ we have:

$$d_{ij} = \left| \tilde{x}_{i,1,\dots j}^{CS} - \tilde{x}_{ij}^{TS} \right|, \tag{6}$$

with $\tilde{x}_{i,1,\dots j}^{CS}$ the cumulative mean of the repeated presentations of the CS (i.e., the perceived CS up until trial $j$) and $\tilde{x}_{ij}^{TS}$ the perceived TS at trial $j$.

As shown in Eq. (2), all of the latent processes in the model are integrated into the generalized associative strength, $g_{ij}$ ($g_{ij} \in [-1, 1]$ for differential conditioning and $g_{ij} \in [0, 1]$ for simple conditioning), with smaller values leading to lower generalized responses and vice versa. The scale of $g_{ij}$ does not correspond to the scale of the observed behaviour (US expectancy VAS, 1–10 range) and thus a scale transformation is necessary. The non-linear sigmoid function with the base rate response and scaling parameters is used to map the latent generalized associative strength to the observed response:

$$\theta_{ij} = A + \frac{K - A}{1 + e^{-(w_{0_i} + w_{1_i} g_{ij})}}, \tag{7}$$

where $A$ and $K$ are the lower and upper limits of the sigmoid function so that $\theta_{ij}$ is commensurable with the measurement scale ($y_{ij} \in [1, 10]$) employed in this study. Therefore, we have chosen

$A = 1$ and $K = 10$. $w0_i$ is the baseline response parameter that governs the response in the absence of CS associative strengths. $w1_i$ is the scaling parameter that governs the mapping between the latent and observed responses (see Supplementary Fig. 3). The new value $\theta_{ij}$ is taken as the mean of the observed response:

$$y_{ij} \sim \mathcal{N}\left(\theta_{ij}, \sigma^2_{m_i}\right), \qquad (8)$$

with $\theta_{ij}$ and $\sigma_{m_i}$ the mean and standard deviation of the normal response distribution.

The parameter $\sigma_{m_i}$ regulates the amount of response noise which depends on the group. To avoid over-fitting and confounding the effect of other parameters, we only asserted group level priors for $\sigma_{m_i}$, which differs between non-learners ($m_i = 1$) and learners ($m_i = 2, 3, 4$). One characteristic of the Non-Learners group is that their final response is completely random and unrelated to the learning or generalization processes. Consequently, two uniform priors are provided for the Non-Learners group and other groups with the criterion that $\sigma_{m_i}$ for the former group is bigger than 1.5, and that $\sigma_{m_i}$ for the remaining groups is smaller than 1.5 (Table 1).

The aforementioned four latent groups can be regarded as four competing models to generate generalized responding. For $m_i = 1$ (i.e., Non-Learners), it depicts that generalized responding is solely determined by the base rate parameter and the response noise parameter and that neither learning nor similarity-based generalization is relevant. Added to $m_i = 1$, $m_i = 2$ (Overgeneralizers) illustrates that learning processes can influence generalized responding, but that generalized responses remain high with at least 70% of the CS response even for most physically different TS. Both $m_i = 3$ (Physical Generalizers) and $m_i = 4$ (Perceptual Generalizers) models assume that individuals learned the CS-US associations to some extent and subsequently generalize them to other stimuli (TS) based on the similarity between the CS(s) and the TS. The distinction between the two groups is whether generalization happens along a physical or perceptual dimension. To allocate participants to latent groups, we employ a categorical distribution to represent the possible outcomes of a random variable that could belong to one of the four groups, with the probability of each category being sampled from the Dirichlet distribution (Table 1).

Individuals are classified into four latent groups based on their response patterns. To avoid erroneous allocation of participants with less certainty about their latent group membership, those who did not satisfy the strict criterion that at least 75% of their posterior samples of the group allocation variable $m_i$ have the same value are being assigned as belonging to an Unknown category. We assessed the sensitivity of our conclusions with respect to more strict and lenient criteria and they did not alter our overall findings (see Supplementary Figs. 4–8 and Supplementary Table 1).

**Simulation study.** Before modelling the data gathered from real experiments, we conducted a simulation study in which we studied the parameter recovery to check if the parameter values are identifiable by the model. We simulated the generalization behaviour of 50 hypothesized participants for each latent group ($m_i = 1, …, 4$) with corresponding data-generating processes and parameter values (total sample size $N = 4 \times 50 = 200$). The experimental structure (i.e., stimuli order, feature estimation, and US presentation) of the simulation is identical to the first participant in Experiment 2.

The parameter values are simulated by the following rules: the learning rate parameter (controlling the amount of learning from a prediction error) $\alpha_i = 0$ for Non-Learners, and $\alpha_i \sim Beta(1,1)$ for

other groups; the generalization rate parameter (controlling the response decay rate given a fixed stimuli distance) $\lambda_i = 0$ for Non-Learners, $\lambda_i \sim N(0.0026, 0.001)T(0, 0.0052)$ for Overgeneralizers, and $\lambda_i \sim N(0.1537, 0.1)T(0.0052, 0.3022)$ for other groups; the response noise parameter $\sigma_1 = 2.5$ for *Non-Learners* and $\sigma_2 = \sigma_3 = \sigma_4 = 0.5$ for other groups; the baseline response parameter $w_{0i} \sim N(0, 5)$ for *Non-Learners* and $w_{0i} \sim N(-2, 1)$ for other groups; the scaling parameter $w_{1i} \sim Gamma(10, 1)$ for all groups.

**Statistical inference.** Parameter estimation is performed by Markov Chain Monte Carlo (MCMC) with the Gibbs sampling method through JAGS[75]. The statistical computing language R[76] and the R package jagsUI[77] are implemented to perform the analysis. Four MCMC chains were run for the model, with 100,000 iterations, 75,000 burn-ins, and a thinning factor of 10 for each chain (i.e., only each 10-th sample was retained). This returns 10,000 samples in total for each parameter. To assess convergence, we use the $\hat{R}$ statistic. MCMC chains are considered as stabilized and have reached the target distribution when the $\hat{R}$ value based on Gelman and Rubin diagnostics[78,79] is close to 1.

The model has a parametric structure that aims at explaining generalized responses. Therefore, it is necessary to ensure that, before drawing any conclusions from parameter estimation, the model fits the data well enough. In this regard, we conducted posterior predictive checks[72,80] to examine the model's fit to the data. The basic idea behind posterior predictive checks is that if the model fits the data, simulated data from the model should resemble the observed data in all possible aspects. In a Bayesian statistical inference framework, data sets are being simulated based on the so-called posterior predictive distribution[80]. If the actual data is unrepresentative of or deviates from the simulated data sets, it can be regarded as a signal that the model does not adequately fit the observed data from the experiment and model modification is needed. For this study, we decided to look at the mean of the observed generalization data (which entails a weak test because it is rather easy to get it right) and a set of quantiles (10%, 30%, 50%, 70%, and 90%) across the test stimulus dimension. Checking deviations between the quantiles of the actual and model-based simulated data allows for a more stringent test because it is a check on the distributional characteristics of the data.

As demonstrated in Fig. 4, in general, the model-based simulated data (i.e., the posterior predictive samples) fit the actual data to a great extent. The posterior predictive of 10% and 90% quantiles show that the model would sometimes generate values outside the observed response scale. This comes from the unconstrained Gaussian likelihood distribution, which is not a problem considering the current research questions. In addition, a satisfactory model fit can also be observed at the individual level (see Supplementary Figs. 9 and 10).

**Reporting summary.** Further information on research design is available in the Nature Portfolio Reporting Summary linked to this article.

## Results

We start the result section with a simulation experiment to test whether we can recover the model's parameters from simulated data. Next, the two experimental data sets are used to demonstrate the ability of the model to identify various generalization-generating mechanisms across individuals. For both simulated and empirical data, the $\hat{R}$ values for most of the parameters are close to 1[78,79], indicating a good convergence for our Bayesian inference algorithm (see Supplementary Figs. 11–36 and Supplementary Tables 2–5).

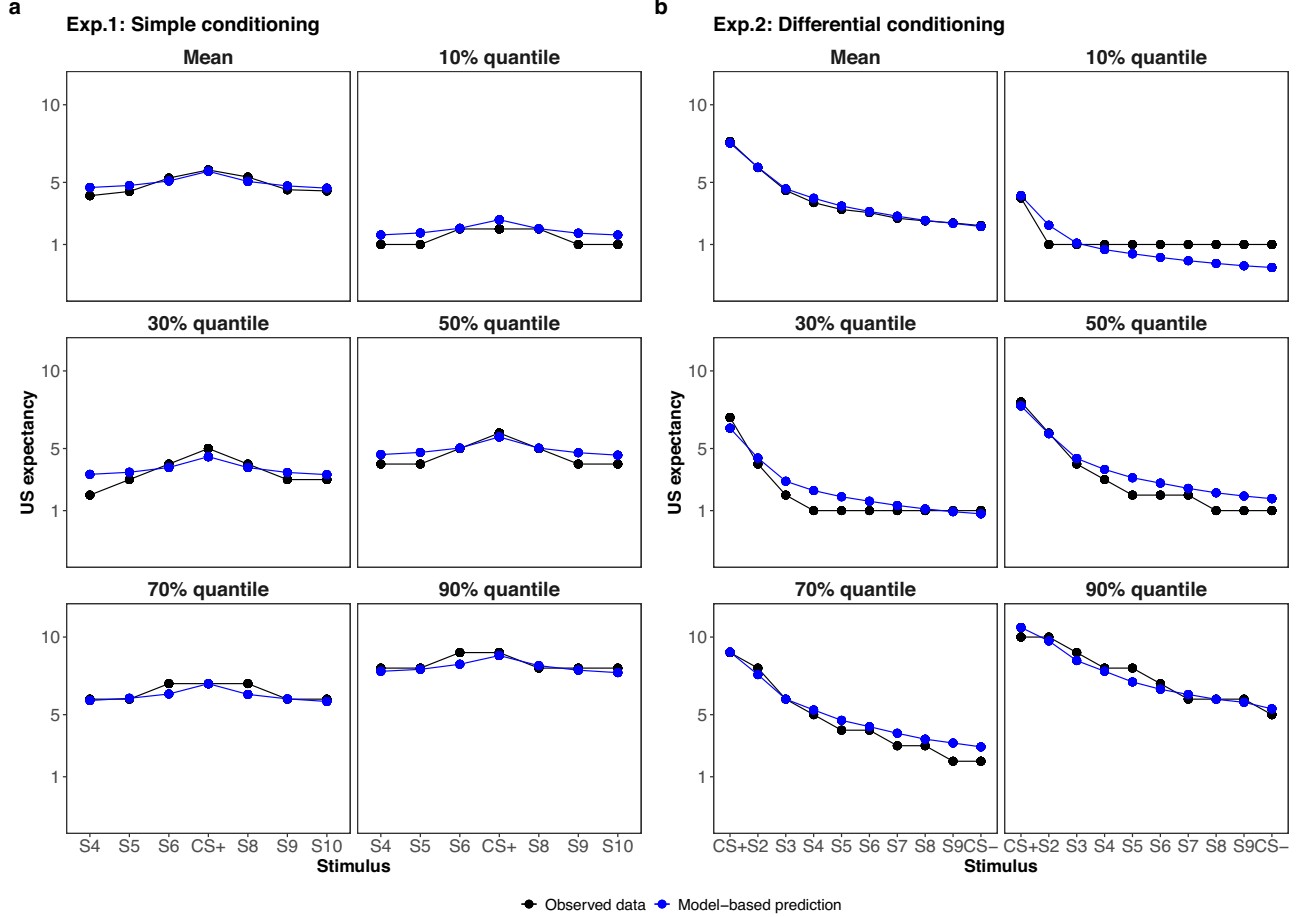

**Fig. 4 Posterior predictive checks.** Comparisons between posterior predictive samples and generalization response data of simple conditioning (panel **a**) and differential conditioning (panel **b**) experiments. The black curve is the mean, 10%, 30%, 50%, 70%, and 90% quantiles of observed responses across different stimuli. The blue curve is the mean, 10%, 30%, 50%, 70%, and 90% quantiles of 5000 replicated data across different stimuli.

**Simulation study**. We generated a total of 200 synthetic participants, with 50 belonging to each of the latent groups, and assigned them varying combinations of parameter values (see the "Method" section for more details). Subsequently, we employed the simulated data to fit the model and investigate the extent of parameter recovery (see Supplementary Fig. 37), which enabled us to establish the relationship between the true parameters - that were utilized to generate the synthetic data of the 200 participants - and the estimated parameters obtained from fitting the model to the simulated data. Our findings indicated that the mixed model is capable of identifying the parameter of interest, specifically, the person-specific generalization rate ($\lambda_i$) and learning rate ($\alpha_i$) simulated by distinct sub-models. Additionally, 95.5 % of synthetic participants are allocated to the correct latent groups (see Supplementary Fig. 38).

In this section, we utilize the synthetic data to provide insights into the underlying generative mechanisms governing the generalization behaviour of the 200 synthetic participants. Furthermore, we demonstrate the implications of overlooking certain mechanisms that are intrinsically linked to the behaviour. Specifically, we investigate how neglecting such mechanisms can lead to biased inferences of the generalization rate parameter, which represents the tendency to generalize.

The group generalization gradient for the simulated data (the average of all individual gradients) is depicted in panel a of Fig. 5. Traditionally, the group gradient has been the primary object of analysis in generalization research. Visually, the group gradient of

the simulated data appears rather flat, indicating a high degree of generalization.

Applying the model to the simulated data allows us to classify the synthetic individuals into four latent groups. By allocating individuals to these latent groups, as illustrated in panel b of Fig. 5, the general patterns of variation in individual responses can be observed. Individuals in Non-Learners and Overgeneralizers groups are the primary contributors for the flatness of the averaged response gradient in panel a of Fig. 5.

In addition, the learning and generalization patterns for each latent group are shown in panels c and d of Fig. 5. Starting with learning, it is demonstrated that Non-Learners do not acquire any associative strengths in the learning trials, whereas the other three groups gain some amount of CS-US associative strengths based on the prediction error in each trial. Non-Learners have never developed a particular differential response to the CSs, which explains why their response gradients are flat across all stimuli. The simulation setting that covers the full range of learning rates for all learner groups explains the comparable learning patterns observed among the groups. Moving on to generalization, Overgeneralizers regard each stimulus to be remarkably similar to the CS, regardless of how physically or perceptually distinct the encountered stimulus is from the CS. This implies that the Overgeneralizers group has an exceptionally strong propensity for generalization and generalizes most of the acquired CS-US associative strengths to every stimulus they encounter. The remaining two groups are both learners and non-

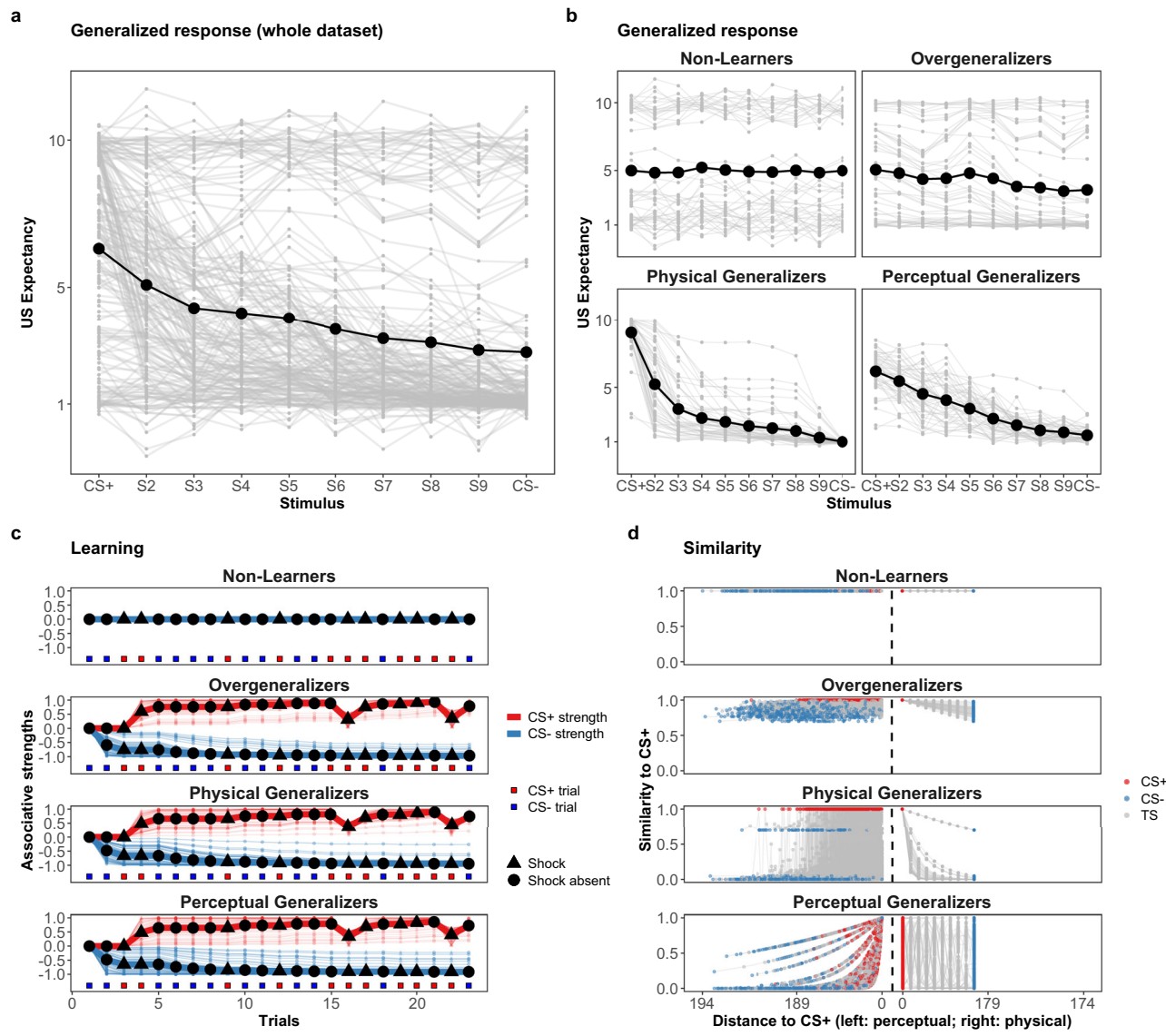

**Fig. 5 Response patterns of 200 synthetic individuals.** Panels **a** and **b**: The dark line connecting with black points is the averaged response gradient of all of the simulated participants ($N = 200$) and of different latent groups across different stimuli. The gray lines are individual response gradients. Panel **c**: The bold and light curves are the averaged and individual CS+ (red) and CS- (blue) associative strengths in the learning stage. Panel **d**: The similarity decay pattern of simulated participants in each latent group. The curves to the left and to the right of the vertical dashed line represents how the similarity to the CS+ decays as a function of perceptual and physical distance, respectively. For *Non-Learners*, similarity to the CS+ remains 100% regardless of the increase of perceptual or physical distance. For *Overgeneralizers*, similarity to the CS+ remains at least 70% with the most physically distant stimulus.

overgeneralizers. Their generalization gradients are less flat compared to Non-Learners and Overgeneralizers. The most notable difference between the two groups is whether or not perceptual variation influences generalization behaviour. As shown in panel d of Fig. 5, similarity to the CS+ declines gradually for the Physical Generalizers group as the encountered stimulus becomes increasingly physically distinct from the CS+. For Perceptual Generalizers, however, the similarity to the CS+ reduces only when they perceive the stimulus to be increasingly distinct from the CS+. Consequently, the strongest response will not always be observed with the CS+, but rather with stimuli that are perceived as the CS+ at a certain time point. In this regard, perceptual errors contribute to the broader generalization gradient observed in the Perceptual Generalizers group compared to the Physical Generalizers groups.

Ignoring other generalization-generating mechanisms can lead to inaccurate estimation of the generalization tendency,

represented by the generalization rate parameter ($\lambda_i$) in our model. To underscore this point, we fitted two simplified models to the simulated data. The first model (Simplified Model 1) omitted the perceptual variability for similarity generalization, while the second model (Simplified Model 2) neglected the learning process. By comparing the parameter estimates obtained from these two models with those derived from our full mixed model, we further demonstrate the potential bias in the estimation of the generalization rate parameter that arises from ignoring related mechanisms.

Figure 6 depicts the $\lambda_i$ recovery results of the two simplified models, which conveys two important insights. First, Simplified Model 1 has difficulties recovering the $\lambda_i$ values of Perceptual Generalizers, whereas it can recover the $\lambda_i$ values of the remaining groups to a great extent. This is because that Simplified Model 1 does not consider the possibility that generalized responding is caused by inaccurately perceiving the CS and TSs, thus it has to

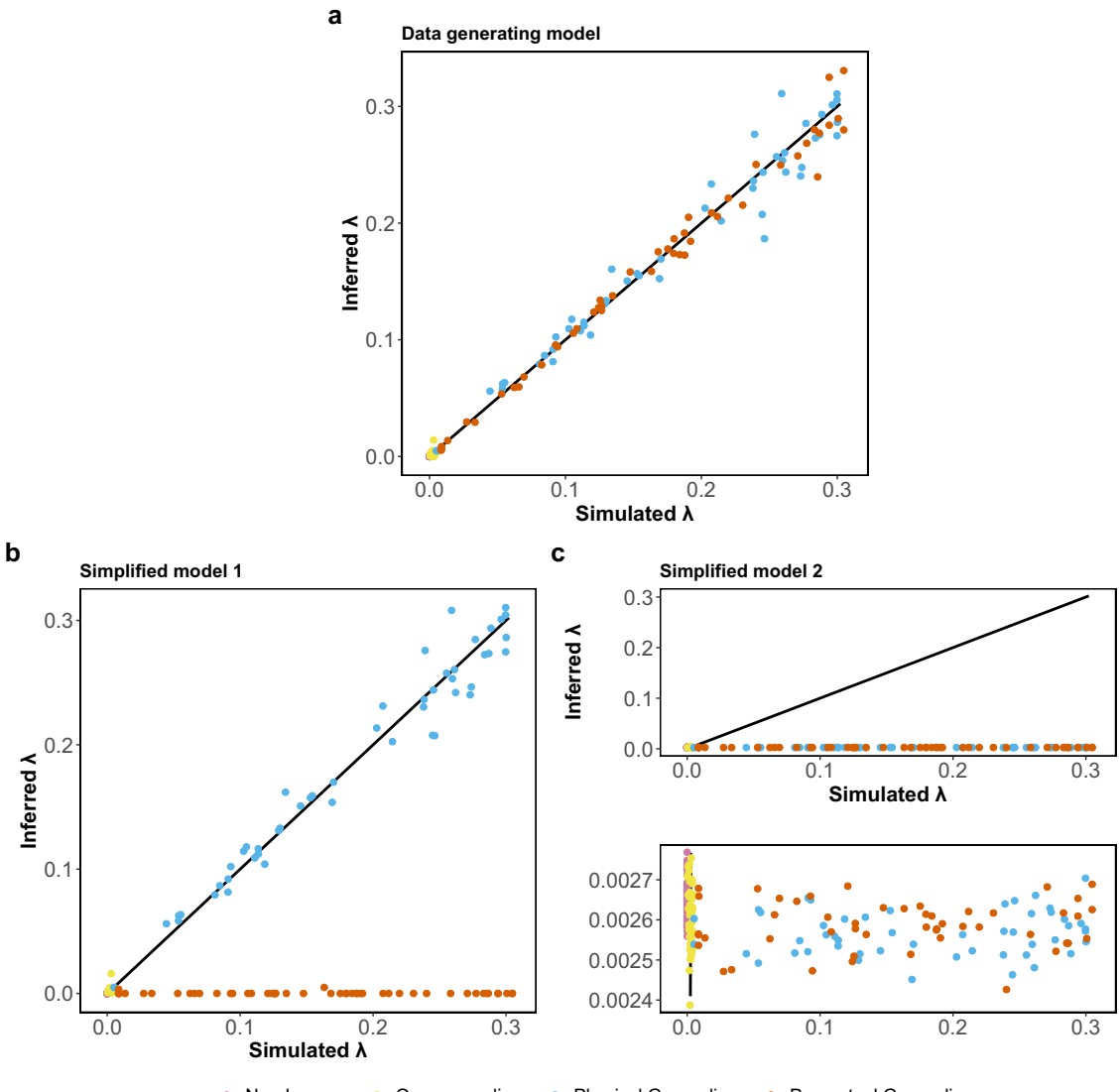

**Fig. 6 Parameter recovery of generalization rates.** The response data is simulated by the mixture model and fitted into the data-generating model (panel **a**) and two simplified models: the model that ignores the perceptual distance for similarity generalization (Simplified Model 1, panel **b**), and the model that disregards the learning process (Simplified Model 2, panel **c**). The second figure in panel **c** provides a zoomed-in view of the *y*-axis from the first figure.

infer a stronger generalization propensity (i.e., smaller $\lambda_i$) to compensate for the generalized responses associated with perceptual errors. In the simulation, individuals in the Perceptual Generalizers group are those who generalize their responses based on the perceptual distance between the CS and TS and thus their $\lambda_i$ would not be recoverable by Simplified Model 1. Second, Simplified Model 2 fails to recover most of the $\lambda_i$ values and the inferred values are all biased toward smaller values. In accordance with the assumptions for learners ($m_i = 2, 3, 4$), learning is a time-dependent process, therefore CS-US associations vary across trials. Consequently, the same generalization rate can result in distinct responses for the same individual at various time points depending on their current learned CS-US association. Simplified Model 2, without considering CS-US learning, has to infer a stronger generalization propensity (i.e., smaller $\lambda_i$) to compensate for the indistinguishable responses associated with weak associative strengths. Consequently, if the impact of learning and perception on generalization behaviour is not taken into account, there is a risk of bias in inferences about generalization tendencies.

**Experimental study**. In this section, we will present the outcomes of the computational model analysis on the data from the two experiments. To evaluate the model fit, we conducted posterior predictive checks, and the results exhibited a noteworthy level of prediction accuracy with various quantiles (Fig. 4).

Figure 7 shows the posterior distributions for the group membership probability (Panel a), and group mean (Panel b) and person-specific (Panel c) learning rate parameters $\alpha_i$ and generalization rate parameters $\lambda_i$, which are the two parameters that modulate the two key processes in the model. With Bayesian modelling, the parameter value is represented by a probability distribution. Therefore, the uncertainty of the parameter estimates is shown with a 95% credible interval (95% CI). In general, the narrower the interval, the less uncertainty we have about a parameter's value (given both the prior knowledge and the observed data).

The posterior probability that participants are allocated to Non-Learners is much higher in the first experiment (95% CI [0.21,0.48], 35% of participants) compared with the second (95% CI [0.03,0.22], 10% of participants), suggesting our model is

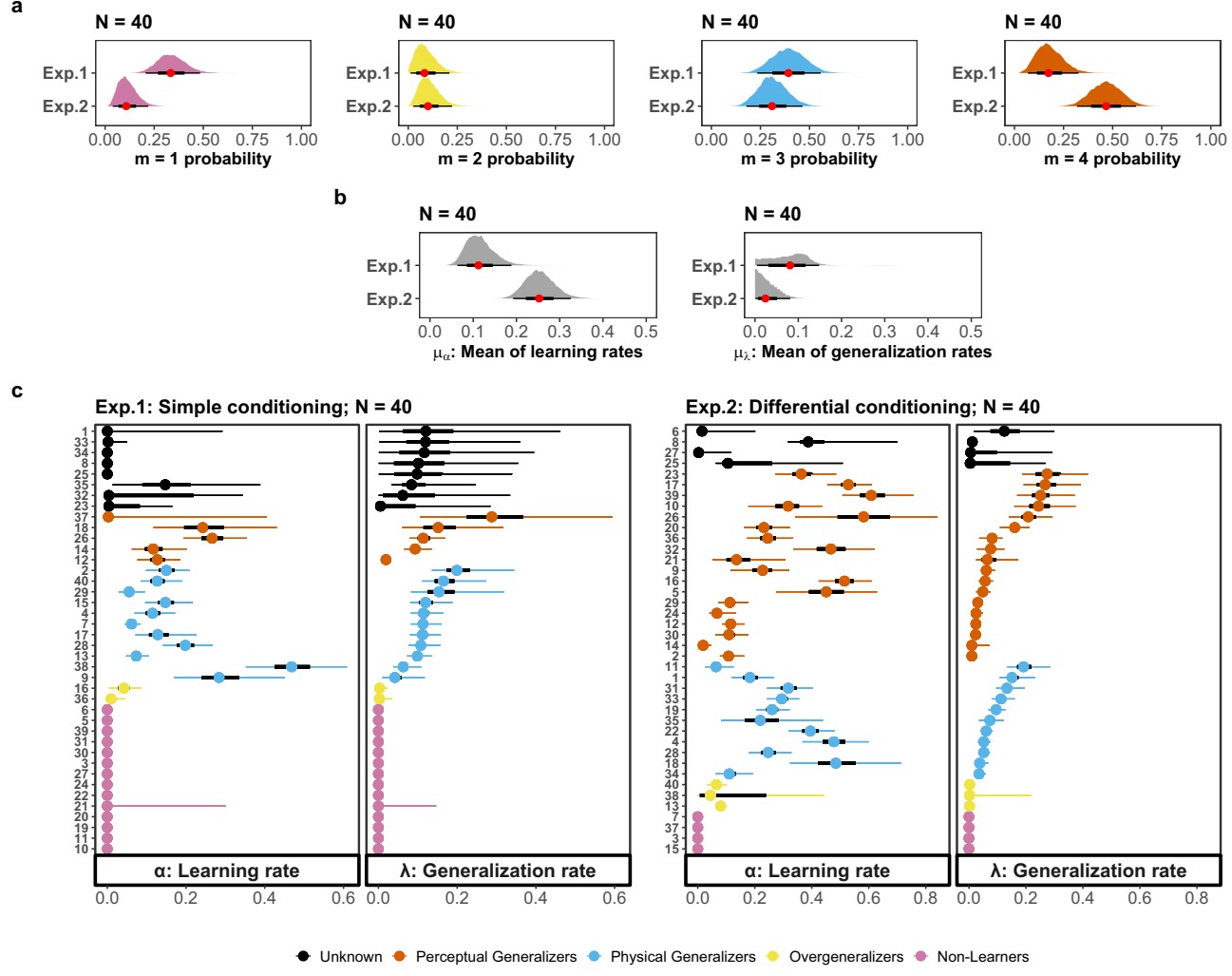

**Fig. 7 Parameter estimates of learning rates and generalization rates.** Parameter estimates with 95% credible interval. Panel **a**: Posterior samples of the group membership variable of each latent group. Panel **b**: Posterior samples of the group mean learning rate $\mu_\alpha$ and generalization rate $\mu_\lambda$. Panel **c**: Posterior samples of the person-specific learning rate $\alpha_i$ and generalization rate $\lambda_i$.

capable of picking up an experimental manipulation targeting this specific latent process. Furthermore, we found a similar probability to observe Overgeneralizers in both experiments (Experiment 1: 95% CI [0.01,0.21], 5% of participants; Experiment 2: 95% CI [0.03,0.22], 7.5% of participants), and Physical Generalizers (Experiment 1: 95% CI [0.23,0.56], 27.5% of participants; Experiment 2: 95% CI [0.18,0.46], 27.5% of participants). The probability of being assigned to the Perceptual Generalizers group is much higher in the second experiment (95% CI [0.32,0.62], 12.5% of participants) compared with the first (95% CI [0.07,0.33], 45% of participants). Based on the posterior samples of the group membership variable $m_i$, which ranges from integer values 1 to 4, participants are assigned to different groups. To address estimation uncertainty, we adopted a strict criterion that requires at least 75% of the $m_i$ samples to have the same value; otherwise, the participant is designated as Unknown.

For the individuals who are effectively assigned to one of the four latent groups, their estimated $\alpha_i$ and $\lambda_i$ values can be interpreted in a meaningful manner. The greater the $\alpha_i$ value, the more learning will occur when the outcome of a stimulus differs from what was anticipated. The greater the $\lambda_i$ value, the greater the response decay at a particular distance between the CS and a TS. Both parameters contribute to the steepness of a

generalization gradient. Consider two participants in Experiment 2 displayed in panel c of Fig. 7, Participant 11 and Participant 4. Participant 11 has a greater $\lambda_i$ and a smaller $\alpha_i$ than Participant 4. Given a fixed distance, Participant 4 may generalize more learning to TS. However, with a greater learning rate $\alpha_i$, his or her acquired associative strength is much higher, and combined results in a net steeper response gradient.

The posterior for the mean of learning rates $\mu_\alpha$ is larger in Experiment 2 (95% CI [0.19,0.32]) compared with Experiment 1 (95% CI [0.06,0.19]). This shows that, as expected, compared to the first experiment (simple conditioning), individuals in the second experiment (differential conditioning) generally better picked up the CS-US associations. However, there is a notable overlap between the two posterior distributions for the mean generalization rates $\mu_\lambda$, which makes it difficult to distinguish between the group-level estimates for the generalization tendency in the two experiments. At the individual level, $\alpha_i$ and $\lambda_i$ are constrained to 0 for the Non-Learners. Overgeneralizers in both experiments tend to have small values of $\alpha_i$ (Experiment 1: 95% CI [0,0.0785]; Experiment 2: 95% CI [0.0011,0.3688]) and $\lambda_i$ (Experiment 1: 95% CI [0.0001,0.026]; Experiment 2: 95% CI [0.0005,0.1167]). A combination of low and high $\alpha_i$ and $\lambda_i$ values can be observed in both the Physical Generalizers and Perceptual Generalizers groups.

Our results revealed that participants had difficulty forming the CS-US association in a simple conditioning paradigm with a lower reinforcement rate (50% of CS+ are followed by a US). Specifically, we observed lower learning rates and a higher proportion of participants (35%) classified as Non-Learners, for whom there was insufficient evidence of learning. In contrast, a differential conditioning paradigm with a higher reinforcement rate (83% of CS+ are followed by a US) fostered stronger learning, as indicated by higher learning rates and a lower proportion of participants (10%) classified as Non-Learners. These findings suggest that regarding learning, our model can accurately attribute experimental manipulations to the intended latent process. The observation of flat response gradients in Non-Learners aligns with previous research investigating the impact of learning paradigms on response gradients[17–21]. Moreover, neglecting to distinguish Non-Learners can lead to bias in the estimation of latent generalization tendency, as demonstrated in the panel c of Fig. 6.

Assuming learning took place, the model explores the extent to which learned CS-US associations are generalized to different TSs based on their (physical or perceptual) distances to the CS. When an individual's generalized response remained at least 70% of the response to the CS when confronted with the most physically distinct stimulus (distance = 68.62 mm), the person was classified as Overgeneralizers. In both data sets, there is only a small proportion of participants being allocated to the group of Overgeneralizers (Experiment 1: 5%; Experiment 2: 7.5%). These individuals have an exceptionally small generalization rate parameter $\lambda_i$ and respond similarly to a wide range of TSs.

The remaining participants demonstrate a reasonable response decay as stimulus distance increases. To investigate the nature of this response decay, we further utilized the mixture model to examine whether individual differences in perception accounted for the observed variation in generalized responding (Perceptual Generalizers) or if generalized responses were unaffected by perception and thus solely attributed to the physical similarity (Physical Generalizers). In the first data set, there are more Physical Generalizers (27.5%) than Perceptual Generalizers (12.5%), whereas, similar to learning, the reverse pattern is found in the second data set, where Perceptual Generalizers are the majority (45%) compared to Physical Generalizers (27.5%). The inverse pattern between experiments was unexpected but may be due to sampling bias where there are far fewer individuals in the first data set classified to these two groups than in the second (40% versus 72.5%). Comparing the response gradients of these subgroups (Figs. 8, 9) reveals that, as expected, Perceptual Generalizers exhibit broader response gradients due to increased perceptual variability, which in turn leads to greater generalized responding[22–26]. These behavioural differences are not indicative of differences in the extent of a latent generalization tendency but rather stem from differences in stimulus perception. Furthermore, neglecting to account for perceptual variability as an intrinsic source of generalization behaviour, as exemplified in Panel b of Fig. 6, could lead to biased inferences concerning the latent generalization tendency, akin to disregarding the accounting learning process.

## Discussion

In this research, we used Bayesian computational modelling combined with multi-source data to distinguish among diverse mechanistic processes and to draw more insightful conclusions regarding inter-individual differences in human generalization behaviour. Specifically, the learning and similarity-based generalization processes are specified as mathematical functions to connect with data gathered from fear conditioning, stimulus

generalization, and stimulus perception tasks. This contrasts favorably with the prevailing approach, which focuses only on the group-level generalization gradient and a single data-generation mechanism (the latent generalization mechanism). Our findings highlight how similar patterns in generalization behaviour can be explained by different mechanisms or how different patterns can be explained by similar mechanisms.

In our model, generalized behaviour is determined by several psychologically meaningful parameters. If a person is identified as neither in the Non-Learners nor Overgeneralizers group, the value for his or her generalization rate parameter $\lambda_i$ reflects his propensity to generalize as the decreasing (transferring) proportion of response given the acquired associative strength (modulated by the learning rate parameter $\alpha_i$) and the perceptual (Perceptual Generalizers) or physical (Physical Generalizers) stimuli distance. Consequently, two participants may have the same propensity for generalization (i.e., the same $\lambda_i$ values) yet radically different flatness of response gradients due to their distinct learning and perceptual abilities.

The four latent groups provide a theoretical framework that not only highlights the impact of diverse mechanisms on individual generalization behaviour but also holds promising clinical implications. Previous findings associating generalization-related psychopathologies with learning[30–32] and generalization mechanisms[1,2,4–6] underscore the importance of identifying Non-Learners and Overgeneralizers in clinical practice. Given that trait anxiety is typically low among healthy individuals[37,38], it stands to reason that the prevalence of Overgeneralizers would be low in a healthy sample, as observed in the present study. Future research could aim to investigate whether the prevalence of this latent group increases in a clinical sample or whether membership in this latent group has any predictive value for treatment outcomes. The detection of Perceptual Generalizers is also noteworthy, particularly when considered alongside previous research indicating that anxiety patients exhibit deficits in perceptual discrimination after fear learning[33,34]. Therefore, it is essential to carefully consider the contribution of perceptual variability (Fig. 1) to generalization behaviour in individuals, whether from clinical or theoretical perspectives.

To obtain a comprehensive understanding of the latent generalization mechanism, it is crucial to assimilate all diverse sources that exert an impact on generalization behaviour. By doing so, the distinct contributions of each source can be disentangled, leading to the derivation of a more meaningful generalization rate parameter value. Disregarding behaviour-related mechanisms, such as learning and perception, which are integral components in the generalization process, can engender biased inferences of the mechanism. Our simulation study has unequivocally shown that neglecting these mechanisms imposes a considerable bias in the extent to which the observed behaviour is attributed to a latent generalization mechanism. Consequently, such distorted inferences of regarding a latent generalization process potentially undermine the validity of subsequent analyses (e.g., relating it to personality traits). Alternatively, if the generalization behaviour is of interest, it remains important to consider all pertinent processes and mechanisms to understand how such behaviour emerged.

Understanding the cognitive mechanisms underlying psychiatric disorders is essential for developing effective treatments and personalized protocols that target the underlying causes. Computational methods have gained prominence in exploring these mechanisms[81–83], particularly through parameterizing specific psychological processes such as learning[84,85]. This approach enables researchers to identify the etiology of psychiatric symptoms, which is necessary for classifying individuals along symptomatic dimensions[86]. Generalization-related psychiatric disorders often co-

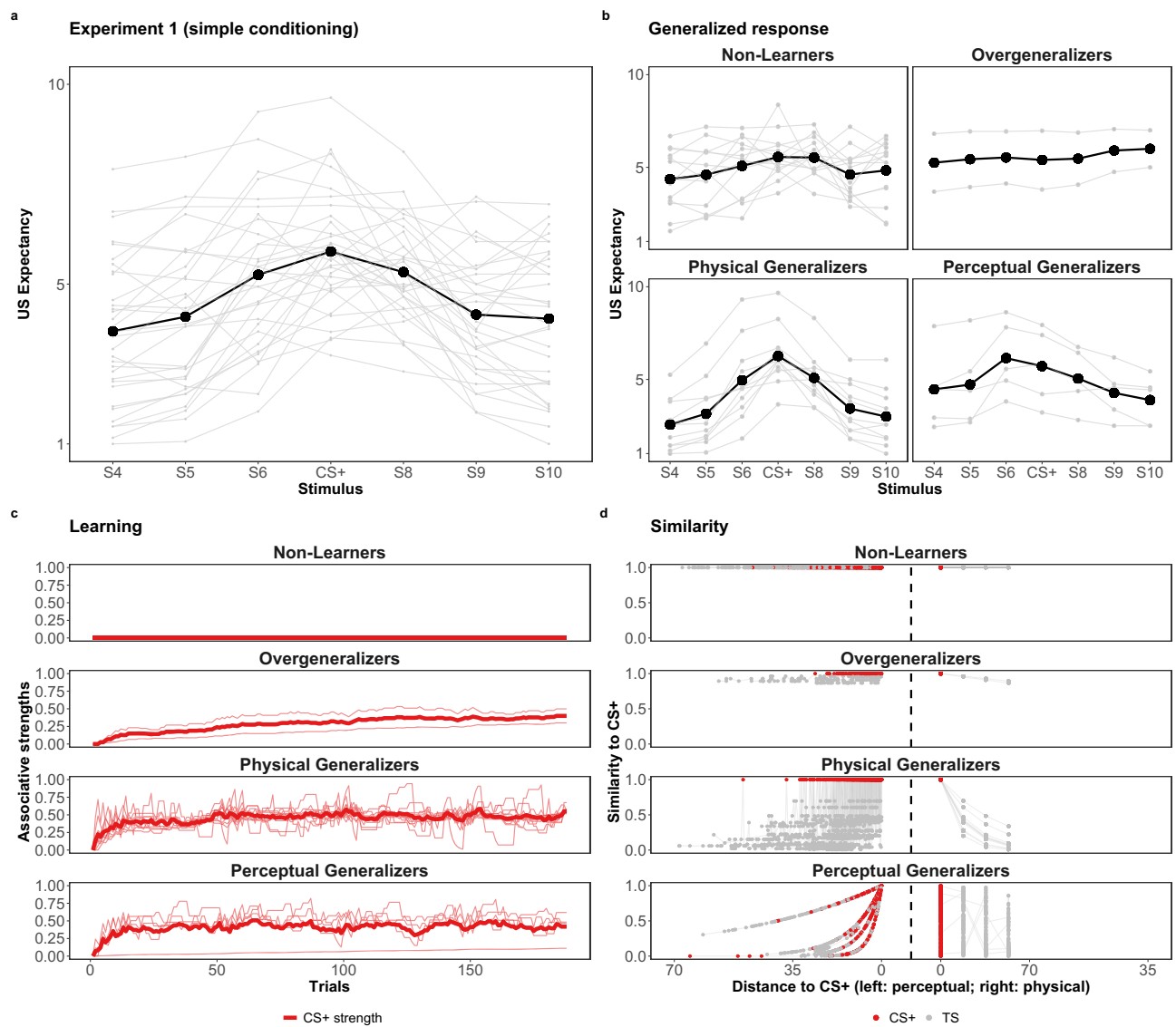

**Fig. 8 Response patterns of Experiment 1.** The observed generalization responses, learning and CS+ similarity decay patterns of Experiment 1 (simple conditioning). Panels **a** and **b**: The dark line connecting with black points is the averaged response gradient of all of the participants (N = 40) and of different latent groups (except for Unknown) across different stimuli. The gray lines are individual response gradients. Panel **c**: The bold and light curves are the averaged and individual CS+ associative strengths with the estimated learning rate (50% quantile) parameter $\alpha_i$. Panel **d**: The individual similarity decay pattern with the estimated generalization rate (50% quantile) parameter $\lambda_i$. The curves to the left and to the right of the vertical dashed line represents how the similarity to the CS+ decays as a function of perceptual and physical distance, respectively. For *Non-Learners*, similarity to the CS+ remains 100% regardless of the increase of perceptual or physical distance. For *Overgeneralizers*, similarity to the CS+ remains at least 70% with the most physically distant stimulus.

occur and display substantial heterogeneity, underscoring the need for diagnostic tools and more effective treatments[87]. One promising approach is the incorporation of meaningful parameters as behavioural readouts in computational models, providing a fundamental framework for investigating how generalization behaviour is formed by different mechanisms. These cognitively interpretable parameters can serve as valuable psychomarkers (as opposed to biomarkers), behavioural endophenotypes, and treatment outcome predictors in clinical settings, which can be correlated with brain activations or personality traits[88]. By capturing learning and generalization parameters in the proposed model, researchers can develop informative indicators and classifiers for generalization-related psychiatric symptoms in future research endeavors. Considering all pertinent processes and mechanisms can help us gain a deeper understanding of how generalization behaviour is formed and how it relates to psychiatric symptoms.

**Future directions**. Future research on developing more sophisticated models of generalization behaviour should consider additional fundamental cognitive processes that may directly impact these phenomena. For instance, attentional processes, which allow for the selection and prioritization of relevant information, have been highlighted as a crucial modulator of learning and generalization[89,90]. Additionally, rule-based generalization based on specific relational differences between stimulus features has been shown to occur alongside distance-based generalization[91,92]. Recent research has also explored the coexistence of these two types of generalization processes and their interaction in producing generalization behaviour[16,24]. In addition, future research should aim to conduct more rigorous experimental studies that manipulate specific psychological processes. This will enable us to determine how much the model can accurately detect the manipulation of the targeted latent process. Moreover, to improve the

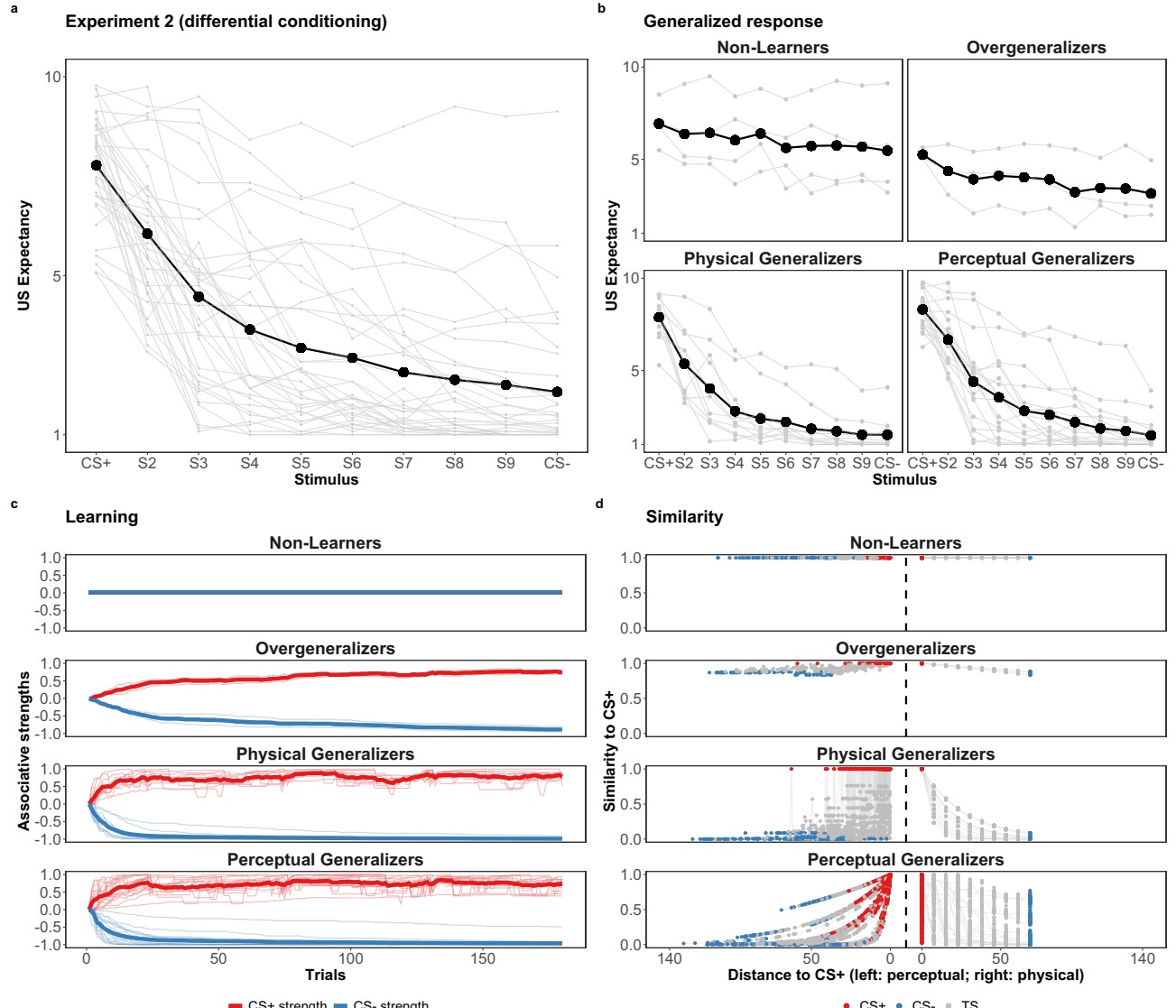

**Fig. 9 Response patterns of Experiment 2.** The observed generalization responses, learning and CS+ similarity decay patterns of Experiment 2 (differential conditioning). Panels **a** and **b**: The dark line connecting with black points is the averaged response gradient of all of the participants ($N = 40$) and of different latent groups (except for *Unknown*) across different stimuli. The gray lines are individual response gradients. Panel **c**: The bold and light curves are the averaged and individual CS+ (red) and CS- (blue) associative strengths with the estimated learning rate (50% quantile) parameter $\alpha_i$. Panel **d**: The individual similarity decay pattern with the estimated generalization rate (50% quantile) parameter $\lambda_i$. The curves to the left and to the right of the vertical dashed line represents how the similarity to the CS+ decays as a function of perceptual and physical distance, respectively. For Non-Learners, similarity to the CS+ remains 100% regardless of the increase of perceptual or physical distance. For Overgeneralizers, similarity to the CS+ remains at least 70% with the most physically distant stimulus.

reliability and validity of our model, future research should concentrate on implementing it using various stimulus sets both in pre-clinical and clinical populations.

The Bayesian approach renders an additional layer to construct assumptions through the prior distributions. To reflect the absence of information about parameters, we only specified ambiguous and weakly informative priors to the hyperparameters in this study. However, having informative priors is important for capturing valuable theoretical and empirical information about the parameters[93]. For example, we can construct priors for the learning rate parameter $\alpha_i$ in simple conditioning paradigms with small values if similar results are systematically replicated. Such prior constructions represent the current knowledge that people tend to learn poorly in such context. The belief updating process for both data generating functions and parameter priors can also

draw great interests for clinicians. On the one hand, the same observed symptom (e.g., anxiety or fear generalization) can be broken down into different mechanisms that potentially require different therapeutic interventions, allowing for the development of tailored treatments. The treatment performance, on the other hand, can be examined by looking at how the prior distribution of the targeted latent parameter evolved. In our model, for instance, the prior value of the learning rate parameter $\alpha_i$ will increase as a result of an effective treatment aimed at reducing generalization behaviour induced by problematic learning.

**Limitations**. The construction of perceptual distance in our model did not incorporate the latest theories on memory and perception. Instead, we employed size estimation data to directly depict TS perception and formulated CS memory as a cumulative average of

the size estimation in CS trials. By adopting this approach, we essentially equated CS memory with encoding precision, overlooking potential variations in retrieval processes. Nevertheless, it is important to acknowledge that this simplified methodology might not fully capture the intricate complexities inherent in the underlying processes. For example, with the recent advancement of human perception research, the possibility that the perceptual system operates according to Bayes' rule has been widely discussed[42–48]. Under this framework, perception contains probabilistic representations that incorporate a likelihood function and prior perceptual knowledge. In such instances, the trial-based single percept representation we provided to the model may be too simplistic to accurately portray how individuals represent the presented stimulus (and its associated uncertainty). Likewise, inter-individual differences during encoding, retrieval or mechanisms acting hereupon may cause substantial biases in memory that can greatly diverge from the objective[94,95]. Furthermore, differences in memory retrieval have been linked to differences in fear generalization patterns[96]. The advantage of the adopted generative model and use of multi-source data is that insights from other fields could be implemented within the existing framework by altering or extending the model and the type of collected data (e.g., memory data). Another limitation of the present research is that our conclusions are based solely on self-report responses. Future studies could overcome this limitation by incorporating multiple response channels, such as physiological and neuronal responses, to investigate generalization behaviour. Previous studies have demonstrated the influence of perceptual variability on startle eyeblink responses, indicating the potential for generalizability of our findings[22,25,97]. In addition, while the current model has been only utilized in the context of fear conditioning paradigms with one-dimensional stimuli, it would be beneficial for future research to investigate the generalizability of the modelling conclusions to other learning types, stimuli with higher levels of complexity and multiple dimensions, and samples.

## Conclusions

Our results emphasize computational (generative) modelling's adaptability for integrating advanced research from other fields into generalization, as well as its ability to unify disparate theories of generalization mechanisms through model comparison and parameter estimation. The current model assumes a basic framework of how humans update CS learning with prediction error and how they exponentially generalize their response to novel stimuli based on either perceptual or physical distances between CS and TS. To advance our understanding of the mechanisms underlying human generalization behaviour, it is essential for future research to extend the current computational framework in parallel with developments in other fields of psychology.

Shepard[12] defined generalization as psychology's first law because it enables us to adapt to the diverse contexts encountered in daily life. Yet, like many other behaviours, generalization is the result of a variety of cognitive and perceptual processes. The true generalization mechanism, if it ever exists, can be discovered only after excluding all other mechanisms that could result in the same behaviour.

## Data availability

The raw and processed data for the two experiments in this study[98] can be accessed at the following Open Science Framework (OSF) repository: https://osf.io/sxjak/.

## Code availability

The code for the computational model and analysis, as well as supplementary information with additional information about the model and results, can be found at the same repository as the data: https://osf.io/sxjak/. The Bayesian sampling is conducted with JAGS (version 4.3.1), and the post-sampling analysis and visualization are conducted with R (version 4.1.1).

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

## Acknowledgements

K.Y. is supported by an FWO research project (co-PI: J.Z., G079520N). J.Z. is a Postdoctoral Research Fellow of the Research Foundation Flanders (FWO, 12P8623N) and received funding from a Special Research Funds (FWO, 1500620N). The research leading to the results reported in this paper was also supported in part by the Research Fund of KU Leuven (C14/19/054) and the resources and services used in this work were provided by the VSC (Flemish Supercomputer Center), funded by FWO and the Flemish Government.

## Author contributions

K.Y. conceptualized and performed the analysis, visualized and interpreted the results, and drafted the manuscript under F.T., W.V., and J.Z.'s supervision. F.T., W.V., and J.Z. critically reviewed and revised the manuscript. J.Z. obtained the research funding for the current project. All authors approved the final version of the article.

## Competing interests

The authors declare no competing interests.
