## [Peer Review File · Communications Psychology]

6th Mar 23

Dear Mr Yu,

Thank you for your patience during the peer-review process.

We're very sorry for the much delayed decision on your manuscript, which came about as a result of the holiday period in December 2022 (which delayed reviewer assignment) and subsequent reviewer unavailability.

Your manuscript titled "Multiple pathways to widespread fears: Disentangling idiosyncratic fear generalization mechanisms using computational modeling" has now been seen by 3 reviewers, and I include their comments at the end of this message. They find your work of interest, but raised some important points. We are interested in the possibility of publishing your study in *Communications Psychology*, but would like to consider your responses to these concerns and assess a revised manuscript before we make a final decision on publication.

We therefore invite you to revise and resubmit your manuscript, along with a point-by-point response to the reviewers. Please highlight all changes in the manuscript text file.

Editorially, we consider the following issues key:

- The reviewers, especially Reviewer #1 highlight instances in which the results do not unambiguously support your interpretation. You must demonstrate clearly how the conclusions arise from this pattern of results. Any remaining caveats must be discussed transparently in the Discussion section under the subheading "Limitations".
- The reviewers are concerned about some analysis choices; in revision, we expect you to provide a compelling rationale for your analysis decisions in response to each individual point, and where necessary demonstrate that alternative analyses do not alter the results.
- Reviewer #2 notes deviations from the preregistration. Please revise the manuscript to address this issue as per our guidelines (<https://www.nature.com/commspsychol/submit/preregistration>): "We ask that authors indicate at the time of submission if any of the work reported in their manuscript was preregistered. If there was no preregistration of any study, this must be declared in the Methods section for transparency. Conversely, if any of the reported studies was preregistered, authors must provide an active link to the preregistration in the Methods section and state the date of preregistration. Authors must disclose all deviations from the preregistered protocol and explain the rationale for deviation (e.g., flaw, feasibility, suboptimality). In cases of deviation from the preregistered analysis plan for reasons other than fundamental flaw or feasibility, the originally planned analyses must also be reported."
- Finally, we ask you to improve the presentation of the model in the manuscript to address relevant reviewer concerns, and ensure that the model code as openly available online is sufficiently well documented and can be accessed without barriers (please see our code sharing guidelines below).

Please use the following link to submit your revised manuscript, point-by-point response to the referees' comments (which should be in a separate document to any cover letter) and the completed checklist:

[link redacted]

We understand that due to the current global situation, the time required for revision may be longer than usual. We would appreciate it if you could keep us informed about an estimated timescale for resubmission, to facilitate our planning. Of course, if you are unable to estimate, we are happy to accommodate necessary extensions nevertheless.

Please do not hesitate to contact me if you have any questions or would like to discuss these revisions further. We look forward to seeing the revised manuscript and thank you for the opportunity to review your work.

Best regards,

Xiaoqing Hu

Xiaoqing Hu, PhD
Editorial Board Member
Communications Psychology
orcid.org/0000-0001-8112-9700

EDITORIAL POLICIES AND FORMATTING

Editorial Policy: Policy requirements (Download the link to your computer as a PDF.)

Furthermore, please align your manuscript with our format requirements, which are summarized on the following checklist:

Communications Psychology formatting checklist

and also in our style and formatting guide <https://www.nature.com/documents/commsj-psychol-style-formatting-guide-accept.pdf>>Communications Psychology formatting guide .

* **CODE AVAILABILITY:** All Communications Psychology manuscripts must include a section titled "Code Availability" at the end of the methods section. In the event of publication, we require that the custom analysis code supporting your conclusions is made available in a publicly accessible repository; at publication, we ask you to choose a repository that provides a DOI for the code; the link to the repository and the DOI will need to be included in the Code Availability statement. Publication as Supplementary Information will not suffice. We ask you to prepare code at this stage, to avoid delays later on in the process.

* **DATA AVAILABILITY:**
All Communications Psychology manuscripts must include a section titled "Data Availability" at the end of the Methods section or main text (if no Methods). More information on this policy, is available at <http://www.nature.com/authors/policies/data/data-availability-statements-data-citations.pdf>><http://www.nature.com/authors/policies/data/data-availability-statements-data-citations.pdf>.

At a minimum the Data availability statement must explain how the data can be obtained and whether there are any restrictions on data sharing. Communications Psychology strongly endorses open sharing of data. If you do make your data openly available, please include in the statement:

We recommend submitting the data to discipline-specific, community-recognized repositories, where possible and a list of recommended repositories is provided at <http://www.nature.com/sdata/policies/repositories>><http://www.nature.com/sdata/policies/repositories>.

If a community resource is unavailable, data can be submitted to generalist repositories such as <https://figshare.com/>>figshare or <http://datadryad.org/>>Dryad

Digital Repository

. Please provide a unique identifier for the data (for example a DOI or a permanent URL) in the data availability statement, if possible. If the repository does not provide identifiers, we encourage authors to supply the search terms that will return the data. For data that have been obtained from publicly available sources, please provide a URL and the specific data product name in the data availability statement. Data with a DOI should be further cited in the methods reference section.

REVIEWERS' EXPERTISE:

Reviewer #1 computational psychiatry, (fear) learning
Reviewer #2 computational psychiatry, (fear) learning
Reviewer #3 computational psychiatry, modelling

REVIEWERS' COMMENTS:

Reviewer #1 (Remarks to the Author):

The manuscript employs Bayesian computational modelling to examine the fear generalization mechanism. Using the observed data in two fear conditioning experiments and combining multiple data source in a Bayesian multilevel mixture model, the authors found that the generalization process during learning can be broken down into four distinct mechanisms, namely non-learners, overgeneralizes, physical generalizers, and perceptual generalizers.

The study has the following strength: 1) revealed the existence of multiple generalization mechanisms instead of assuming one generalization mechanism; 2) took account of different mechanisms influencing mental representations, which ultimately influence the generalization process, instead of using mono-source behavioral data; 3) account for individual differences in modeling, instead of merely focusing on the differences at the group level.

The manuscript was well written with clear explanations on the statistical analyses and results. The findings have potential implications for the development of tailored treatments in the clinical context. Upon clarifications of certain findings and the text from the authors, I am confident to recommend the publication of this manuscript in your prestigious journal.

Here are some questions I have on this manuscript:

1. Introduction line 5: While generalized learning are reduced in individuals with autism, there is insufficient evidence to date to show that problematic generalization “lead to the development of “ autism. Please revise the sentence.
2. Introduction Paragraph 5: “ CR to test stimuli were strong when they were falsely perceived as CS (even during CS trials)”. “even during CS trials” need a reference.
3. Methods: The total number of participants in experiments 1 and 2 is $40 + 43 = 83$. Why does the final dataset comprise 40 participants only?

4. I assume that it is a parametric Bayesian model. Could you give the rationale for choosing this model over the non-parametric one in your study?
5. Figure 4 Panel D: the text does not match the description of the findings in the figure. The curves to the right of the vertical dashed line represents the similarity to the CS+ decay as a function of physical distance, not perceptual.
6. Figure 4 Panel C: The learning pattern does not really differentiate well among the three latent groups (overgeneralizes, physical generalizers and perceptual generalizers) and the pattern look the same to me in these three groups. Is it a common/normal finding and could you please comment further in the text?
7. Relatedly, there is no red curve in the learning pattern. Please revise the description in figure 4c.
8. The authors did not explain why one group of participants would generalize TS with CS based on the physical distance between TS and CS, and the other would exhibit generalization because of changes in inter-stimulus distances. From a gestalt perspective, the brain may process these two cognitive factors in a holistic manner and generate ultimate behavior based on a combination of both factors, which vary in weight in different trials. If the authors think such a categorical classification outweighs a continuous classification (with different weights of physical generalization and perceptual generalization for different individuals), the authors should address the limits of the latter.
9. Some limitations should be highlighted: As it is rightly stated by the authors, it is important to consider a variety of cognitive and perceptual processes that are pertinent to the generalization learning. Working memory and attention functioning are also implicated in the generalization and the learning processes (likely they influence a_i and g_{ij} in the model) but are not included or accounted for in your model. Also, the physiological measure (the startle responses) are not included in the computational model. There are some of the limitations in your study.
10. I find the findings of the simulation model very interesting to both researchers and clinicians. Specifically, 35% of the participants in a simple conditioning task are non-learners while 10% in a differential conditioning are non-learners. Statistically, it explains why in traditional experimental study we often need to exclude some non-learners in the analyses. Likewise, it also highlights the individual differences in terms of the generalized learning of fear in the clinical population.

Reviewer #2 (Remarks to the Author):

In this paper, the authors present a new model for generalization behavior in a conditioning paradigm. They argue that generalization behavior can be the result of several distinct (mental) processes such as imperfect associative learning or perceptual discrimination. Furthermore, they explain that modeling can also be a valuable approach for generalization behavior because it allows to better take into account individual differences.

The paper presents the results of a simulation study and two experiments, both with 40 participants. The first experiment involved a simple fear conditioning procedure with a moderate reinforcement rate, whereas the second experiment used a differential fear conditioning procedure (CS+ and CS-) with high reinforcement rate for the CS+. Differences

over experiments in the estimated model parameters were in accordance with the model assumptions about these parameters.

Overall, I found this paper very interesting to read. I'm not a modeling expert, but the details of the model seemed well-specified and model decisions seemed sensible. I also agree with the basic arguments in favor of a more dynamic model of the processes underlying generalization behavior. I am also convinced that such a model may be of great use for the generalization literature because a more precise analysis of generalization behavior (that takes into account individual-level variables) can be valuable for practice and theory.

That said, there were a couple of things that came to mind while I was reading the paper that the authors might use to further improve the paper.

- For me, it remains unclear why the authors attempt to categorize participants into four groups. Is it not more useful to estimate how data (at the participant and group-level) relate to specific processes (e.g., as the authors do with the learning and generalization parameter)? Some more explanation might be useful.

- It's always important not to oversell the results of modeling experiments. I think the authors could be more sensitive to the fact that modeling results cannot provide strong evidence for specific processes as these processes are self-defined and self-selected (e.g., the authors decided not to model motivation-related processes). This is especially important when the model is presented as allowing a 'mechanism-specific differential diagnosis in generalization-related psychiatric disorders'. It is important to note that the value of the model will need to be tested in reference to specific goals that one might have (e.g., to predict or influence behavior). This can be explained a bit more (how can the model help achieve certain goals and what research is important to test this?). I do like the fact that the authors note that they will update the model based on new insights such as related to predictive coding (but this is now described in the supplementary material). In the paper, it wasn't always clear how recent insights are taken into account and how the behavior was thought to be related to specific processes (also: are all processes defined at the mental level?).

- It's a bit strange that the authors did not pursue the analyses they had preregistered. Why not? Is there nonetheless some information to be found online about the results of these analyses?

Reviewer #3 (Remarks to the Author):

In this work the authors used computational modeling to better understand the possible underlying mechanisms of fear generalization on an individual level. Such understanding can help to promote personalized approaches in research and clinical realms, by shifting the focus from groups to individuals. While potentially interesting, there are a few major concerns that attenuate my enthusiasm and should be considered if a decision to revise is made:

Major:

- My main concern is that this report is heavily focused on the computational aspect of this work, and the depth and details of the modeling work are often hard to follow. Importantly, such focus on the computational aspect drives the attention away from the important implications to psychology in general, and to psychopathologies in particular. While clearly the authors performed a large amount of analytical work, I'd suggest that a major revision is performed to make this manuscript a better fit for a psychology journal. Below I listed a few specific examples how a stronger link to the clinical field can be made.
- The model focuses on 2 parameters: learning rate and generalization rate. The report would benefit from a greater elaboration on the relevance to humans, e.g., personality traits and/or clinical characteristics in specific patient populations.
- The presented model classifies participants into four distinct groups. It would be important to provide a detailed explanation how these groups were chosen, how does such classification can fit empirical data in the generalization literature, and if there are other groups that could be considered in future related work.
- Similarly, this work tests 3 specific mechanisms: learning, perception and generalization. It would be important to have a clearer explanation of these 3 mechanisms, how they are related to the existing literature and if there are any other mechanisms that should be considered in future work (see also suggestion below re context processing).
- One major part of this work is the empirical part, where two generalization datasets have been collected. It would be important to include a separate section that discusses in greater detail these studies, including demographics, task design, and empirical data. Specifically, it would be helpful to present the behavior data as usually presented in the generalization literature, with level of responding to each class of stimuli (CS-, CS+, various generalization stimuli).

Minor:

- Are the implications of this work specific to fear generalization, or relevant to other generalization/transfer learnings? Authors might want to discuss the generalizability of this work.
- Authors can also discuss other possible cognitive mechanisms that can contribute to generalization but were not studied here. For instance, content processing, which is altered in some psychopathologies, and its related pattern separation and pattern completion processes, could be discussed. Can the reported model be used to better understand how context processing affects generalization patterns?
- In the generalization literatures, the novel "similar" stimuli are usually termed "generalization stimuli" and not "test stimuli" as in the current report. Making the terminology more consistent with the literature would be helpful.
- In the introduction, the first and second limitations seem to be very overlapping. To understand the various possible mechanisms that can drive generalization patterns (e.g., perceptual discrimination; limitation #1), one needs to utilize multi-source data (e.g., self-reported similarity judgement; limitation #2). I'd suggest that these limitations are combined, or alternatively, the distinction is clarified.
- "An abundance of empirical evidence has shown that individual differences in learning and perception may account for the diversity in generalization behavior" — In light of the motivation of this work to study individual differences, I suggest elaborating on the existing

literature that studied these individual differences. Further, in the discussion, the authors could explain how/if the conclusions from this work fit this existing literature.

Rebuttal letter

April 24 2023

Reviewer 1

1.

Introduction line 5: While generalized learning are reduced in individuals with autism, there is insufficient evidence to date to show that problematic generalization “lead to the development of “ autism. Please revise the sentence.

We agree with the Reviewer’s suggestion that the phrasing "lead to the development of" may not be optimal in the absence of clear causal evidence. Therefore, we revised the sentence in the Introduction line 5 as follows:

"Maladaptive generalization behavior is associated with several psychopathologies, such as anxiety disorder (Dymond et al., 2015; Lissek et al., 2014), autism (Plaisted, 2001), and obsessive-compulsive disorder (Kaczkurkin and Lissek, 2013; Rouhani et al., 2019)."

2.

Introduction Paragraph 5: “ CR to test stimuli were strong when they were falsely perceived as CS (even during CS trials)”. “even during CS trials” need a reference.

We have added the following references to this sentence:

"Conditioned responses to test stimuli were strong when they were falsely perceived as the CS while attenuated when the stimulus was perceived as different from the CS (even during CS trials) (Struyf et al., 2015; Zaman et al., 2019; Zaman et al., 2020; Zaman et al., 2022). This inflated the inferred extent of fear generalization as response gradients were much narrower after accounting for these perceptual errors (Zaman, Struyf, et al., 2021)".

3.

Methods: The total number of participants in experiments 1 and 2 is $40 + 43 = 83$. Why does the final dataset comprise 40 participants only?

The pre-registered sample size of 40 only included participants who finished the experiment. For Experiment 1, since the data was lost for 3 participants due to technical issues we had to recruit a total of 43 subjects to reach the pre-registered sample size of 40 completed data sets. This is reported in the *Method* section:

"To reach the pre-registered sample size of 40 complete data sets in both experiments we had to recruit 43 and 40 participants, respectively, as three participants in the first experiment did not complete the task due to technical problems."

4.

I assume that it is a parametric Bayesian model. Could you give the rationale for choosing this model over the non-parametric one in your study?

We chose to use a parametric Bayesian model for our study because we wanted to investigate whether learning, perceptual variability, and generalization tendency are all important sources in human generalization behavior, and if a computational model can distinguish the effects of these mechanisms. With the parametric model, we can mathematically describe how generalized responses emerge, providing insights into the underlying cognitive and perceptual processes. Additionally, the parametric model allows us to quantify the strength and direction of each mechanism, providing more specific and interpretable results.

Non-parametric (or semi-parametric) models (such as neural networks) do not make strong assumptions about the functional form of the process being modeled, but they estimate the relationship between the input and output based on the observed data. While this approach can be useful in cases where the underlying process is complex or nonlinear, it may not provide the level of interpretability that a parametric model can offer. Moreover, the parametric Bayesian model enabled us to incorporate prior knowledge or beliefs about the mechanisms and parameters of interest, which is a critical advantage over non-parametric models. By incorporating prior knowledge, we can reduce the influence of noise or variability in the data and obtain more robust estimates of the parameters of interest. In addition, prior knowledge can facilitate the integration of new studies with past findings of certain processes. While non-parametric models can be useful in certain situations (most commonly when there is an abundance of data, which is often not the case in human learning experiments), they cannot offer the same level of interpretability, specificity, and ability to incorporate prior knowledge as the parametric Bayesian model. Therefore, for our study, we concluded that the parametric Bayesian model was the most appropriate choice to achieve our research goals and to provide a deeper understanding of human generalization behavior.

5.

Figure 4 Panel D: the text does not match the description of the findings in the figure. The curves to the right of the vertical dashed line

represents the similarity to the CS+ decay as a function of physical distance, not perceptual.

We thank the Reviewer for noticing and corrected the mistake in the caption of Figure 3 (which was previously Figure 4):

"Panel D: The similarity decay pattern of simulated participants in each latent group. The curves to the right and to the left of the vertical dashed line represents how the similarity to the CS+ decays as a function of **physical and perceptual** distance, respectively."

6.

Figure 4 Panel C: The learning pattern does not really differentiate well among the three latent groups (overgeneralizes, physical generalizers and perceptual generalizers) and the pattern look the same to me in these three groups. Is it a common/normal finding and could you please comment further in the text?

The similar learning patterns among these three latent groups is due to the fact that our simulation covered the full range of possible learning rate values, and we did not specifically model different types of learning patterns between these three subgroups. This decision was made because we did not expect these subgroups to differ in learning but rather in other aspects. Thus, the similar learning patterns observed in the simulations among these three groups is expected. We only expect to see deviations regarding learning patterns in the Non-Learners group for whom learning has never been established. Note that for the experimental data (see Figure 6 and Figure 7), some small differences in learning patterns among the three groups can be seen. To avoid confusion, we added a sentence in the *Description of the simulated response patterns* section (page 6):

"The simulation setting that covers the full range of learning rates for all learner groups explains the comparable learning patterns observed among the groups. "

7.

Relatedly, there is no red curve in the learning pattern. Please revise the description in figure 4c.

We apologize for the mistake and have adjusted Figure 3 (previously Figure 4). Now, the red curve indicates CS+ learning, and the blue curve indicates CS- learning.

"Panel C: The bold and light curves are the averaged and individual CS+ (red) and CS- (blue) associative strengths in the learning stage."

8.

The authors did not explain why one group of participants would generalize TS with CS based on the physical distance between TS and CS, and the other would exhibit generalization because of changes in inter-stimulus distances. From a gestalt perspective, the brain may process these two cognitive factors in a holistic manner and generate ultimate behavior based on a combination of both factors, which vary in weight in different trials. If the authors think such a categorical classification outweighs a continuous classification (with different weights of physical generalization and perceptual generalization for different individuals), the authors should address the limits of the latter.

In our model, the *Physical Generalizers* group is defined as situations where perceptual variability does not play a significant role in determining generalization behavior. This outcome can occur in two possible scenarios: Firstly, when perceptual variability is negligible or very small. Secondly, when the degree of perceptual variability is unrelated to the observed generalization behavior, indicating that perceptual variability does not significantly influence the way the individual generalizes from one stimulus to another. As the Reviewer suggested, it is also possible to investigate the different weights of two distances (physical and perceptual) for different individuals and assess the contribution of perceptual variability based on the weighting parameter. However, the current study aims to provide a straightforward generalization tendency index that can be easily interpreted. The current model framework incorporates a generalization rate parameter that quantifies the degree of response decay for a particular associative strength as a function of the distance between stimuli. In other words, the generalization rate parameter reflects how much the response to a certain stimulus weakens as the stimulus becomes further away from the original stimulus. Introducing a weighting parameter would make the generalization rate parameter more difficult to interpret and less comparable between individuals. Therefore, we did not consider this option in the current study.

We added a few sentences in the manuscript to increase clarity.

1. In the *Introduction* section (page 2), we emphasized the difference of two perceptual assumptions:

"However, previous research on generalization has largely ignored the potential impact of inter-individual differences, often by either equating perception to the physical dimension (making perception the same for everyone) or employing psychophysical mapping methods such as multidimensional scaling (Shepard,

1957, 1987) to derive a mental representation of encountered stimuli. These psychophysical mapping methods mostly assume a universal (and invariant) psychological space, with stimulus representations in an identical constellation for all humans, rendering them unable to account for inter- and intra-individual variations in perception, the impact of learning on perception (Gibson, 1953; Razran, 1955; Sagi, 2011; Watanabe et al., 2001; Zhang et al., 2019) and the impact of both on generalized responses (Raviv et al., 2022; Schroyen et al., 2015; Zaman et al., 2017)."

2. In the *Introduction* section (page 5), we also emphasized the distinction between the two groups:

"Fourth, we incorporate a mixture framework (Bartlema et al., 2014; Lee, 2011; Winsberg and De Soete, 1993) that allows us to allocate individuals into potential clinical-relevant subgroups based on the malfunctioning of certain latent processes. The specific pattern of generalized behavior can be the result of (1) problematic learning (i.e., *Non-Learners*), and when learning occurred, (2) the extreme generalization tendency (i.e., *Overgeneralizers*), or (3) a similarity-based generalization process where differences in stimulus perception influence the extent of stimulus similarity (i.e., *Perceptual Generalizers*), or (4) a similarity-based generalization process where perceptual variability does not impact stimulus similarity (i.e., *Physical Generalizers*). This tree-like structure naturally leads to four groups."

3. Similarly, in the *Result summary* section (page 11), we added:

"The remaining participants demonstrate a reasonable response decay as stimulus distance increases. To investigate the nature of this response decay, we further utilized the mixture model to examine whether individual differences in perception accounted for the observed variation in generalized responding (*Perceptual Generalizers*) or if generalized responses were unaffected by perception and thus solely attributed to the physical similarity (*Physical Generalizers*)."

4. In the *Introduction* section (page 5), we emphasized that the model aims to provide a more interpretable index for generalization tendency:

"The proposed model not only examines the impact of distinct latent processes on generalized responses but also identifies potential clinical subgroups among participants by accounting for specificities in their latent processes. Furthermore, it parameterizes the latent similarity-based generalization process (in straightforward index for generalization propensity) in such a manner that it enables the comparison between individuals despite their differences in learning and perceptual sensitivity. Focusing on the behavior-generating process, the model investigates how different presumptive mechanisms combined (i.e. learning, perception, generalization) influence the generalization behavior patterns of different individuals. "

5. Also in the *Discussion* section (page 14), we added:

"In our model, generalized behavior is determined by several psychologically meaningful parameters. If a person is identified as neither in the *Non-Learners* nor *Overgeneralizers* group, the value for his or her generalization rate parameter λ_i reflects his propensity to generalize as the decreasing (transferring) proportion of response given the acquired associative strength (modulated by the learning rate parameter α_i) and the perceptual (*Perceptual Generalizers*) or physical (*Physical Generalizers*) stimuli distance, both of which can vary across time points within and between individuals. Consequently, two participants may have the same propensity for generalization (i.e., the same λ_i values) yet radically different flatness of response gradients due to their distinct learning and perceptual abilities. "

9.

Some limitations should be highlighted: As it is rightly stated by the authors, it is important to consider a variety of cognitive and perceptual processes that are pertinent to the generalization learning. Working memory and attention functioning are also implicated in the generalization and the learning processes (likely they influence a_i and g_{ij} in the model) but are not included or accounted for in your model. Also, the physiological measure (the startle responses) are not included in the computational model. There are some of the limitations in your study.

We added two subsections *Future directions* and *Limitations* at the end of *Discussion* section (page 15) to highlight these limitations and the potential future development with relevant cognitive and perceptual theories.

Future directions:

"Future research on developing more sophisticated models of generalization behavior should consider additional fundamental cognitive processes that may directly impact these phenomena. For instance, attentional processes, which allow for the selection and prioritization of relevant information, have been highlighted as a crucial modulator of learning and generalization (Bedard and Song, 2013; Dayan et al., 2000). Additionally, rule-based generalization based on specific relational differences between stimulus features has been shown to occur alongside distance-based generalization (Dunsmoor and Murphy, 2015; Nosofsky and Zaki, 2002). Recent research has also explored the coexistence of these two types of generalization processes and their interaction in producing generalization behavior (Schlegelmilch et al., 2020; Zaman et al., 2022). In addition, future research should aim to conduct more rigorous experimental

studies that manipulate specific psychological processes. This will enable us to determine how much the model can accurately detect the manipulation of the targeted latent process. Moreover, to improve the reliability and validity of our model, future research should concentrate on implementing it using various stimulus sets both in pre-clinical and clinical populations.

The Bayesian approach renders an additional layer to construct assumptions through the prior distributions. To reflect the absence of information about parameters, we only specified ambiguous and weakly informative priors to the hyperparameters in this study. However, having informative priors is important for capturing valuable theoretical and empirical information about the parameters (Vanpaemel and Lee, 2012). For example, we can construct priors for the learning rate parameter α_i in simple conditioning paradigms with small values if similar results are systematically replicated. Such prior constructions represent the current knowledge that people tend to learn poorly in such context. The belief updating process for both data generating functions and parameter priors can also draw great interests for clinicians. On the one hand, the same observed symptom (e.g., anxiety or fear generalization) can be broken down into different mechanisms that potentially require different therapeutic interventions, allowing for the development of tailored treatments. The treatment performance, on the other hand, can be examined by looking at how the prior distribution of the targeted latent parameter evolved. In our model, for instance, the prior value of the learning rate parameter α_i will increase as a result of an effective treatment aimed at reducing generalization behavior induced by problematic learning."

Limitations:

"The formation of perceptual distance in our model did not account for potential general patterns over time and idiosyncrasies in human perception and memory. Rather, we directly represented TS perception using size estimation data and modeled CS memory as a moving mean of the size estimation in CS trials. This approach equated CS memory with encoding precision, while neglecting potential differences in retrieval. However, this approach may oversimplify the actual complexity of the underlying processes. For example, with the recent advancement of human perception research, the possibility that the perceptual system operates according to Bayes' rule has been widely discussed (Colombo and Seriès, 2012; Gross, 2020; Knill and Alexandre, 2004; Petzschner et al., 2015; Press et al., 2020; Press and Yon, 2019; Weiss et al., 2002). Under this framework, perception contains probabilistic representations that incorporate a likelihood function and prior perceptual knowledge. In such instances, the trial-based single percept representation we provided to the model may be too simplistic to accurately portray how individuals represent the presented stimulus (and its associated uncertainty). Likewise, inter-individual differences during encoding, retrieval or mechanisms acting hereupon may cause substantial biases in memory that can greatly diverge from the objective (Brady and Alvarez, 2011; Luck and Vogel, 2013). Furthermore, differences in memory retrieval have been linked to differences in fear generalization patterns (Zenses

et al., 2021). The advantage of the adopted generative model and use of multi-source data is that insights from other fields could be implemented within the existing framework by altering or extending the model and the type of collected data (e.g., memory data). Another limitation of the present research is that our conclusions are based solely on self-report responses. Future studies could overcome this limitation by incorporating multiple response channels, such as physiological and neuronal responses, to investigate generalization behavior. In addition, while the current model has been only utilized in the context of fear conditioning paradigms with one-dimensional stimuli, it would be beneficial for future research to investigate the generalizability of the modeling conclusions to other learning types, stimuli with higher levels of complexity and multiple dimensions, and samples."

10.

I find the findings of the simulation model very interesting to both researchers and clinicians. Specifically, 35% of the participants in a simple conditioning task are non-learners while 10% in a differential conditioning are non-learners. Statistically, it explains why in traditional experimental study we often need to exclude some non-learners in the analyses. Likewise, it also highlights the individual differences in terms of the generalized learning of fear in the clinical population.

We appreciate that the reviewer found our modeling results interesting and would like to thank the Reviewer for the helpful suggestions

Reviewer 2

1.

For me, it remains unclear why the authors attempt to categorize participants into four groups. Is it not more useful to estimate how data (at the participant and group-level) relate to specific processes (e.g., as the authors do with the learning and generalization parameter)? Some more explanation might be useful.

We apologize for any confusion regarding this matter. In this study, our goal is to investigate whether generalization behavior can be decomposed into learning, perceptual variability, and generalization tendency. To achieve this, we built a parametric computational model that incorporates parameters that embed these psychological processes. Instead of subsequently conducting post-hoc analysis on posterior samples to determine whether someone belongs to a potential clinical and/or theoretical relevant subgroup (i.e., a non-learner, overgeneralizer), which embeds more subjectivity, we decided to combine the parameter estimation for specific processes at once with a mixture model structure that enables us to identify individuals belonging to these subgroups based on their estimates parameter values. Thus our model serves two aims: (1) estimating individual parameters (and their uncertainty) relating to distinct processes in order to scrutinize their impact on generalized responses, and (2) allocating participants into potential clinically relevant subgroups based on the posterior parameter distributions. Thus, the purpose of our mixture modeling approach is not to classify individuals into mechanistically distinct groups but merely to label individuals who tend to have some clinically and theoretically relevant particularities regarding these latent processes. We have clarified this in the paper as follows::

1. In the *Introduction* section (page 5), we specified the attempt to allocate participants into potential clinical-relevant subgroups:

"The proposed model not only examines the impact of distinct latent processes on generalized responses but also identifies potential clinical subgroups among participants by accounting for specificities in their latent processes. "

2. In the same page we also emphasized the logic of the mixture modeling framework::

"Our computational model is implemented as a Bayesian multilevel mixture model on the collected multi-source data (i.e., learning, perception, and generalization) and contains four important properties. First, given the complexity of generalization that may emerge from multiple mechanisms, we employ Bayesian statistics to characterize our uncertainty about the parameters as probability distributions in a principled manner (Lee, 2018; Lee and Wagenmakers, 2005).

This allowed us to assess the effect of different psychological processes on human generalization, taking into account the available evidence. Second, the multilevel structure is implemented to account for the ubiquitous quantitative individual differences by inferring parameter values from both the individual and group levels (Lee, 2011; Lee and Vanpaemel, 2008; Okada and Lee, 2016; Scheibehenne and Pachur, 2015). Third, in the model, generalization behavior is not restricted to a single generating process but could emerge from a several different latent processes situated at different levels (e.g., learning, generalization). Fourth, we incorporate a mixture framework (Bartlema et al., 2014; Lee, 2011; Winsberg and De Soete, 1993) that allows us to allocate individuals into potential clinical-relevant subgroups based on the malfunctioning of certain latent processes. The specific pattern of generalized behavior can be the result of (1) problematic learning (i.e., *Non-Learners*), and when learning occurred, (2) the extreme generalization tendency (i.e., *Overgeneralizers*), or (3) a similarity-based generalization process where perceptual variability does not impact stimulus similarity (i.e., *Physical Generalizers*), or (4) a similarity-based generalization process where differences in stimulus perception influence the extent of stimulus similarity (i.e., *Perceptual Generalizers*). This tree-like structure naturally leads to four groups."

3. In the *Discussion* section (page 14), we discussed the theoretical and clinical relevance of the defined latent groups:

"The four latent groups provide a theoretical framework that not only highlights the impact of diverse mechanisms on individual generalization behavior but also holds promising clinical implications. Previous findings associating anxiety disorders with problematic learning processes (Hermann et al., 2002; Jovanovic et al., 2010; Lissek et al., 2005) and a tendency to overgeneralize (Dymond et al., 2015; Kaczurkin and Lissek, 2013; Lissek et al., 2014; Plaisted, 2001; Rouhani et al., 2019) underscore the importance of identifying *Non-Learners* and *Overgeneralizers* in clinical practice. Given that trait anxiety is typically low among healthy individuals (Dibbets and Evers, 2017; Dibbets et al., 2015), it stands to reason that the prevalence of *Overgeneralizers* would be low in a healthy sample, as observed in the present study. Future research could aim to investigate whether the prevalence of this latent group increases in a clinical sample or whether membership in this latent group has any predictive value for treatment outcomes. The detection of *Perceptual Generalizers* is also noteworthy, particularly when considered alongside previous research indicating that anxiety patients exhibit deficits in perceptual discrimination after fear learning (Duits et al., 2015; Laufer et al., 2016). Therefore, it is essential to carefully consider the contribution of perceptual variability (Figure 1) to generalization behavior in individuals, whether from clinical or theoretical perspectives."

2.

It's always important not to oversell the results of modeling experiments. I think the authors could be more sensitive to the fact that modeling results cannot provide strong evidence for specific processes as these processes are self-defined and self-selected (e.g., the authors decided not to model motivation-related processes). This is especially important when the model is presented as allowing a 'mechanism-specific differential diagnosis in generalization-related psychiatric disorders'. It is important to note that the value of the model will need to be tested in reference to specific goals that one might have (e.g., to predict or influence behavior). This can be explained a bit more (how can the model help achieve certain goals and what research is important to test this?). I do like the fact that the authors note that they will update the model based on new insights such as related to predictive coding (but this is now described in the supplementary material). In the paper, it wasn't always clear how recent insights are taken into account and how the behavior was thought to be related to specific processes (also: are all processes defined at the mental level?).

We agree with the Reviewer that the relevance of the model indeed depends on its ability to pick up experimental manipulation of certain latent processes or the ability to make clinically meaningful predictions. To address this, we have identified three specific steps for our future work. Firstly, we plan to conduct additional experimental studies with more rigorous control for specific psychological processes. This will allow us to determine whether the manipulation of latent processes can be accurately detected by our model. Secondly, we aim to continually expand our model by incorporating insights from relevant domains and theories. By doing so, we can ensure that our model remains up-to-date and relevant in capturing the complexities of generalization. Finally, once our model has been refined to incorporate the most recent insights on the relevant psychological processes, we will undertake clinical translation work to investigate its predictive potential. This will help us determine whether our model can make clinically meaningful predictions. To reflect these future directions, we have added two subsections *Future directions* and *Limitations* in the *Discussion* section:

Future directions:

"Future research on developing more sophisticated models of generalization behavior should consider additional fundamental cognitive processes that may directly impact these phenomena. For instance, attentional processes, which allow for the selection and prioritization of relevant information, have been highlighted as a crucial modulator of learning and generalization (Bedard and Song, 2013; Dayan et al., 2000). Additionally, rule-based generalization based on specific relational differences between stimulus features has been shown to

occur alongside distance-based generalization (Dunsmoor and Murphy, 2015; Nosofsky and Zaki, 2002). Recent research has also explored the coexistence of these two types of generalization processes and their interaction in producing generalization behavior (Schlegelmilch et al., 2020; Zaman et al., 2022). In addition, future research should aim to conduct more rigorous experimental studies that manipulate specific psychological processes. This will enable us to determine how much the model can accurately detect the manipulation of the targeted latent process. Moreover, to improve the reliability and validity of our model, future research should concentrate on implementing it using various stimulus sets both in pre-clinical and clinical populations.

The Bayesian approach renders an additional layer to construct assumptions through the prior distributions. To reflect the absence of information about parameters, we only specified ambiguous and weakly informative priors to the hyperparameters in this study. However, having informative priors is important for capturing valuable theoretical and empirical information about the parameters (Vanpaemel and Lee, 2012). For example, we can construct priors for the learning rate parameter α_i in simple conditioning paradigms with small values if similar results are systematically replicated. Such prior constructions represent the current knowledge that people tend to learn poorly in such context. The belief updating process for both data generating functions and parameter priors can also draw great interests for clinicians. On the one hand, the same observed symptom (e.g., anxiety or fear generalization) can be broken down into different mechanisms that potentially require different therapeutic interventions, allowing for the development of tailored treatments. The treatment performance, on the other hand, can be examined by looking at how the prior distribution of the targeted latent parameter evolved. In our model, for instance, the prior value of the learning rate parameter α_i will increase as a result of an effective treatment aimed at reducing generalization behavior induced by problematic learning.

Limitations:

"The formation of perceptual distance in our model did not account for potential general patterns over time and idiosyncrasies in human perception and memory. Rather, we directly represented TS perception using size estimation data and modeled CS memory as a moving mean of the size estimation in CS trials. This approach equated CS memory with encoding precision while neglecting potential differences in retrieval. In addition, by equating perceptual responses directly to the mental, we oversimplify the actual complexity of the underlying perceptual process. For example, with the recent advancement of human perception research, the possibility that the perceptual system operates according to Bayes' rule has been widely discussed (Colombo and Seriès, 2012; Gross, 2020; Knill and Alexandre, 2004; Petzschner et al., 2015; Press et al., 2020; Press and Yon, 2019; Weiss et al., 2002). Under this framework, perception contains probabilistic representations that incorporate a likelihood function and prior perceptual knowledge. In such instances, the trial-based single percept representation we provided to the model may be too simplistic to

accurately portray how individuals represent the presented stimulus (and its associated uncertainty). Likewise, inter-individual differences during encoding, retrieval or mechanisms acting hereupon may cause substantial biases in memory that can greatly diverge from the objective (Brady and Alvarez, 2011; Luck and Vogel, 2013). Furthermore, differences in memory retrieval have been linked to differences in fear generalization patterns (Zenses et al., 2021). The advantage of the adopted generative model and use of multi-source data is that insights from other fields could be implemented within the existing framework by altering or extending the model and the type of collected data (e.g., memory data). Another limitation of the present research is that our conclusions are based solely on self-report responses. Future studies could overcome this limitation by incorporating multiple response channels, such as physiological and neuronal responses, to investigate generalization behavior. In addition, While the current model has been only utilized in the context of fear conditioning paradigms with one-dimensional stimuli, it would be beneficial for future research to investigate the generalizability of the modeling conclusions to other learning types, stimuli with higher levels of complexity and multiple dimensions, and samples."

3.

It's a bit strange that the authors did not pursue the analyses they had preregistered. Why not? Is there nonetheless some information to be found online about the results of these analyses?

The current work used two unreported data sets which are pre-registered and collected at 2017. In the pre-registered analysis plan, it was proposed to investigate the relationship between perception and generalization response using descriptive statistical analyses (i.e., linear mixed models). However, the current research aims at decomposing generalization behavior into different mechanisms and processes, which requires the formation of a generative model to incorporate the relationship of different mechanisms. The pre-registration that we reported is only for data collection but not for the current study as the research questions are different. Due to this, the current study should be considered as being non-preregistered if hypotheses and analyses are concerned. We have revised the first paragraph of *Method* section to clarify this.

"The two data sets used in the experimental study are taken from two studies that were pre-registered on the Open Science Framework at 2017 (OSF; Experiment 1: <https://osf.io/b4ngs>; Experiment 2: <https://osf.io/t4bzs>). These studies focused on different hypotheses and research questions and used different methodologies than those of the current study, so the plans for descriptive statistical analyses included in the protocol are not relevant for the current study. As the current research questions and corresponding analysis method, which is based on computational modeling, were not included in this, or any other, preregistration protocol, the current study should be considered as being non-

preregistered, as far as hypotheses and analyses are concerned. The raw data for the two experiments, codes for the computational model and analysis, and Supplementary Materials which contains more modeling details, are available at another OSF repository: <https://osf.io/sxjak/>."

Reviewer 3

1.

My main concern is that this report is heavily focused on the computational aspect of this work, and the depth and details of the modeling work are often hard to follow. Importantly, such focus on the computational aspect drives the attention away from the important implications to psychology in general, and to psychopathologies in particular. While clearly the authors performed a large amount of analytical work, I'd suggest that a major revision is performed to make this manuscript a better fit for a psychology journal. Below I listed a few specific examples how a stronger link to the clinical field can be made.

We appreciate that the Reviewer acknowledges the general implications of our work to the field of psychology and to the domain of psychopathologies in specific. We made several adjustments based on the Reviewer's suggestions below to emphasize its clinical relevance.

2.

The model focuses on 2 parameters: learning rate and generalization rate. The report would benefit from a greater elaboration on the relevance to humans, e.g., personality traits and/or clinical characteristics in specific patient populations.

We included some modifications to highlight the utility of parameterized computational models in understanding psychiatric disorders, and to emphasize the potential future use of our model in this context. We also mention previous research on the relationship between learning, perception or generalization and personality traits or patient populations to emphasize their clinical relevance.

1. In the *Introduction* section (page 2), we discussed the clinical relevance of different mechanisms in generalization behavior.

"Yet, the traditional approach predominantly attributes differences in generalized responding to a single generalization mechanism, or in some instances neglects to acknowledge any underlying mechanisms altogether. As a result, the inherent variability of the aforementioned mechanisms and their contribution to the observed behavior is often disregarded (Zaman, Chalkia, et al., 2021). Additionally, it is crucial to acknowledge the clinical relevance of both the learning (Hermann et al., 2002; Jovanovic et al., 2010; Lissek et al., 2005) and perceptual (Corlett et al., 2019; Duits et al., 2015; Laufer et al., 2016; Powers et al., 2017) aspects of generalization, as evidenced by reported differences between healthy individuals and patients. In healthy individuals, evidence has also shown associ-

ations between the level of state anxiety and impaired learning or generalization (Dibbets and Evers, 2017; Dibbets et al., 2015). Hence, a thorough comprehension of (pathological) generalization behavior, encompassing both theoretical and clinical perspectives, necessitates the inclusion of multiple mechanisms in the study. Relying solely on a single cognitive mechanism risks producing incomplete or misleading conclusions, emphasizing the need for a comprehensive approach. "

2. In the *Discussion* section (page 14), we discussed the future potential of the current parameterized model in clinical settings.

"Understanding the cognitive mechanisms underlying psychiatric disorders is essential for developing effective treatments and personalized protocols that target the underlying causes. Computational methods have gained prominence in exploring these mechanisms (Bennett et al., 2019; Frässle et al., 2018; Wang and Krystal, 2014), particularly through parameterizing specific psychological processes such as learning (Maia and Frank, 2011; Mkrтчian et al., 2017). This approach enables researchers to identify the etiology of psychiatric symptoms, which is necessary for classifying individuals along symptomatic dimensions (Wiecki et al., 2015). Generalization-related psychiatric disorders often co-occur and exhibit significant heterogeneity, underscoring the need for diagnostic tools and more effective treatments (Insel et al., 2010). One promising approach is the incorporation of meaningful parameters as behavioral readouts in computational models, providing a fundamental framework for investigating how generalization behavior is formed by different mechanisms. These cognitively interpretable parameters can serve as valuable psychomarkers (as opposed to biomarkers), behavioral endophenotypes, and treatment outcome predictors in clinical settings, which can be correlated with brain activations or personality traits (Stephan and Mathys, 2014). By capturing learning and generalization parameters in the proposed model, researchers can develop informative indicators and classifiers for generalization-related psychiatric symptoms in future research endeavors. Considering all pertinent processes and mechanisms can help gain a deeper understanding of how generalization behavior is formed and how it relates to psychiatric symptoms, ultimately leading to more effective treatments for individuals with these disorders."

3. In the same section, we discussed how neglecting relevant generalization-generating mechanisms can cause potential biased estimations of generalization tendency and thus undermine the validity of subsequent analysis (e.g., relating it to personality traits).

"To obtain a comprehensive understanding of the latent generalization mechanism, it is crucial to assimilate all diverse sources that exert an impact on generalization behavior. By doing so, the distinct contributions of each source can be disentangled, leading to the derivation of a more meaningful generalization rate parameter value. Disregarding behavior-related mechanisms, such

as learning and perception, which are integral components in the generalization process, can engender biased inferences of the mechanism. Our simulation study has unequivocally shown that neglecting these mechanisms imposes a considerable bias in the extent to which the observed behavior is attributed to a latent generalization mechanism. Consequently, such distorted inferences of regarding a latent generalization process potentially undermine the validity of subsequent analyses (e.g., relating it to personality traits). Alternatively, if the generalization behavior is of interest, it remains important to consider all pertinent processes and mechanisms to understand how such behavior emerged."

4. Finally, we included a subsection (page 15) *Future direction* in *Discussion* to highlight the need to test our model with clinical populations in future research:

"Future research on developing more sophisticated models of generalization behavior should consider additional fundamental cognitive processes that may directly impact these phenomena. For instance, attentional processes, which allow for the selection and prioritization of relevant information, have been highlighted as a crucial modulator of learning and generalization (Bedard and Song, 2013; Dayan et al., 2000). Additionally, rule-based generalization based on specific relational differences between stimulus features has been shown to occur alongside distance-based generalization (Dunsmoor and Murphy, 2015; Nosofsky and Zaki, 2002). Recent research has also explored the coexistence of these two types of generalization processes and their interaction in producing generalization behavior (Schlegelmilch et al., 2020; Zaman et al., 2022). In addition, future research should aim to conduct more rigorous experimental studies that manipulate specific psychological processes. This will enable us to determine how much the model can accurately detect the manipulation of the targeted latent process. Moreover, to improve the reliability and validity of our model, future research should concentrate on implementing it using various stimulus sets both in pre-clinical and clinical populations.

The Bayesian approach renders an additional layer to construct assumptions through the prior distributions. To reflect the absence of information about parameters, we only specified ambiguous and weakly informative priors to the hyperparameters in this study. However, having informative priors is important for capturing valuable theoretical and empirical information about the parameters (Vanpaemel and Lee, 2012). For example, we can construct priors for the learning rate parameter α_i in simple conditioning paradigms with small values if similar results are systematically replicated. Such prior constructions represent the current knowledge that people tend to learn poorly in such context. The belief updating process for both data generating functions and parameter priors can also draw great interests for clinicians. On the one hand, the same observed symptom (e.g., anxiety or fear generalization) can be broken down into different mechanisms that potentially require different therapeutic interventions, allowing for the development of tailored treatments. The treatment performance, on the other hand, can be examined by looking at how the prior distribution of the

targeted latent parameter evolved. In our model, for instance, the prior value of the learning rate parameter α_i will increase as a result of an effective treatment aimed at reducing generalization behavior induced by problematic learning."

3.

The presented model classifies participants into four distinct groups. It would be important to provide a detailed explanation how these groups were chosen, how does such classification can fit empirical data in the generalization literature, and if there are other groups that could be considered in future related work.

We agree with the Reviewer and included the following clarifications to better convey the rationale for the presented subgroups.

1. In the *Introduction* section (page 5), we specified the attempt to allocate participants into potential clinical-relevant subgroups:

"The proposed model not only examines the impact of distinct latent processes on generalized responses but also identifies potential clinical subgroups among participants by accounting for specificities in their latent processes. "

2. In the same page we also emphasized the logic of the mixture modeling framework::

"Our computational model is implemented as a Bayesian multilevel mixture model on the collected multi-source data (i.e., learning, perception, and generalization) and contains four important properties. First, given the complexity of generalization that may emerge from multiple mechanisms, we employ Bayesian statistics to characterize our uncertainty about the parameters as probability distributions in a principled manner (Lee, 2018; Lee and Wagenmakers, 2005). This allowed us to assess the effect of different psychological processes on human generalization, taking into account the available evidence. Second, the multilevel structure is implemented to account for the ubiquitous quantitative individual differences by inferring parameter values from both the individual and group levels (Lee, 2011; Lee and Vanpaemel, 2008; Okada and Lee, 2016; Scheibehenne and Pachur, 2015). **Third, in the model, generalization behavior is not restricted to a single generating process but could emerge from a several different latent processes situated at different levels (e.g., learning, generalization).** Fourth, we incorporate a mixture framework (Bartlema et al., 2014; Lee, 2011; Winsberg and De Soete, 1993) that allows us to allocate individuals into potential clinical-relevant subgroups based on the malfunctioning of certain latent processes. The specific pattern of generalized behavior can be the result of (1) problematic learning (i.e., *Non-Learners*), and when learning occurred, (2) the extreme generalization tendency (i.e., *Overgeneralizers*), or

(3) a similarity-based generalization process where perceptual variability does not impact stimulus similarity (i.e., *Physical Generalizers*), or (4) a similarity-based generalization process where differences in stimulus perception influence the extent of stimulus similarity (i.e., *Perceptual Generalizers*). This tree-like structure naturally leads to four groups."

3. In the *Discussion* section (page 14), we discussed the theoretical and clinical relevance of each latent group:

"The four latent groups provide a theoretical framework that not only highlights the impact of diverse mechanisms on individual generalization behavior but also holds promising clinical implications. Previous findings associating generalization-related psychopathologies with learning (Hermann et al., 2002; Jovanovic et al., 2010; Lissek et al., 2005) and generalization mechanisms (Dymond et al., 2015; Kaczkurkin and Lissek, 2013; Lissek et al., 2014; Plaisted, 2001; Rouhani et al., 2019) underscore the importance of identifying *Non-Learners* and *Overgeneralizers* in clinical practice. Given that trait anxiety is typically low among healthy individuals (Dibbets and Evers, 2017; Dibbets et al., 2015), it stands to reason that the prevalence of *Overgeneralizers* would be low in a healthy sample, as observed in the present study. Future research could aim to investigate whether the prevalence of this latent group increases in a clinical sample or whether membership in this latent group has any predictive value for treatment outcomes. The detection of *Perceptual Generalizers* is also noteworthy, particularly when considered alongside previous research indicating that anxiety patients exhibit deficits in perceptual discrimination after fear learning (Duits et al., 2015; Laufer et al., 2016). Therefore, it is essential to carefully consider the contribution of perceptual variability (Figure 1) to generalization behavior in individuals, whether from clinical or theoretical perspectives."

4.

Similarly, this work tests 3 specific mechanisms: learning, perception and generalization. It would be important to have a clearer explanation of these 3 mechanisms, how they are related to the existing literature and if there are any other mechanisms that should be considered in future work (see also suggestion below re context processing).

We thank the Reviewer for the suggestion and rewrote parts of the introduction and discussion to situate them within the existing literature. The fourth paragraph of the *Introduction* section (page 2) aims at explaining how these three mechanisms can influence the final generalization behavior with a discussion of previous empirical research. We have revised the paragraph to make the distinction clearer.

"However, such a single mechanism perspective is not realistic. Generalization can be regarded as a cognitive process whereby previous learning is transferred to newly encountered stimuli based on their similarity to the originally learned stimulus (Shepard, 1957, 1987). Thus, if any aspect of the process - from the initial learning to stimulus perception to the actual transfer of learning - goes awry, it may result in aberrant generalization behavior. For instance, it has been observed that manipulating the experimental learning experiences, such as the number of learning trials, the reinforcement rate, and the learning procedures, exerts a direct influence on generalization gradients (Hanson, 1959; Hearst, 1968; Hull, 1947; Jenkins and Harrison, 1962; Thomas and Switalski, 1966). Without proper learning, organisms are incapable of generating distinct and consistent responses. However, even when CS-US learning is firmly established, generalized responses can still occur as a result of imperfect perceptual discrimination between the CS and newly encountered stimuli (Struyf et al., 2015; Zaman et al., 2019; Zaman, Struyf, et al., 2021; Zaman et al., 2020; Zaman et al., 2022), providing the second potential mechanism for generating flatter or noisy response gradients. In earlier research, the relationship between insufficient perceptual discrimination and generalization behavior has been discussed (Guttman and Kalish, 1956; Kalish, 1958). Recently, researchers have further pointed out the relationship between stochastic perception and generalization behavior. The implication is simple: a generalized response to TSs may not necessarily emerge due to a generalization process but merely emerge due to an inability to perceive the TS as different from the CS. In support of this idea, the high prevalence of problematic stimulus discrimination during a generalization protocol was demonstrated with strong effects on the strength of generalized responding. Conditioned responses to test stimuli were strong when they were falsely perceived as the CS while attenuated when the stimulus was perceived as different from the CS (even during CS trials) (Struyf et al., 2015; Zaman et al., 2019; Zaman et al., 2020; Zaman et al., 2022). This inflated the inferred extent of a latent generalization process as response gradients were much narrower after accounting for perceptual errors (Zaman, Struyf, et al., 2021). Yet, the traditional approach predominantly attributes differences in generalized responding to a single generalization mechanism, or in some instances, neglects to acknowledge any underlying mechanisms altogether. As a result, the inherent variability of the aforementioned mechanisms and their contribution to the observed behavior is often disregarded (Zaman, Chalkia, et al., 2021). Additionally, it is pertinent to note that certain patient populations often exhibit impaired learning (Duits et al., 2015; Hermann et al., 2002; Jovanovic et al., 2010; Lissek et al., 2005), misperceptions (Corlett et al., 2019; Laufer et al., 2016; Powers et al., 2017), and flatter generalization (Dymond et al., 2015; Fraunfelder et al., 2022; Kaczurkin and Lissek, 2013; Lissek et al., 2014; Plaisted, 2001; Rouhani et al., 2019) patterns when compared to healthy controls. Even at the pre-clinical level, variations in these underlying processes have been associated with state anxiety levels (Dibbets and Evers, 2017; Dibbets et al., 2015). This highlights the need to develop tools capable

of identifying and classifying heterogeneous patient population into mechanistic distinct subgroups. Hence, a thorough comprehension of (pathological) generalization behavior, encompassing both theoretical and clinical perspectives, necessitates the inclusion of multiple mechanisms in the study. As indicated by recent research, by relying solely on a single cognitive mechanism, researchers and clinicians are at risk of arriving at incomplete or misleading conclusions, emphasizing the need for a comprehensive approach."

5.

One major part of this work is the empirical part, where two generalization datasets have been collected. It would be important to include a separate section that discusses in greater detail these studies, including demographics, task design, and empirical data. Specifically, it would be helpful to present the behavior data as usually presented in the generalization literature, with level of responding to each class of stimuli (CS-, CS+, various generalization stimuli).

We have provided additional experimental details in the *Introduction* section, in addition to discussing the experimental designs in detail in the *Method* section. This addition serves to enhance the reader's understanding of the study's rationale and methodology from the outset. Furthermore, to better illustrate our findings, we have included generalization gradients in the *Result Discussion* section, in addition to the generalization gradients presented in Figure 1. We hope these additions contribute to the clarity of our study.

1. In the fourth paragraph from the bottom of the *Introduction* section (page 5):

"Before elaborating the computational model further, we first discuss the data that are used to test our model. We used two fear generalization data sets (Experiment 1: $N = 40$, Experiment 2 : $N = 40$). Both experiments employed circles of varying sizes as CS and TS, with a painful electric shock serving as US (for a detailed account, see the *Method* section). During each trial in both the *learning* and *generalization* phases of each experiment, participants had to estimate the stimulus size (i.e., perceptual data) and provide US expectancy ratings (i.e., learning and generalization data) (see Figure 1). Experiment 1 (mean age = 21.8 years, SD = 5.3, 26 females (65%)) used a simple fear conditioning (i.e., one cue preceding a painful US) procedure with a lower reinforcement rate (50%), while in Experiment 2 (mean age = 23.5 years, SD = 8.9, 26 females (60%)), differential fear conditioning was adopted (i.e., two cues, one preceding a painful US, one predictive of the absence of pain) and a higher reinforcement rate (83%). With Experiment 2 compared to 1 differing on aspects that should foster learning, we can examine if the model can capture different patterns of generalized behavior of participants under very different learning experiences.

"

2. In the *Result discussion* session (page 11-13), Figure 6 and Figure 7 display the generalization gradients.

3. In the *Method* session, we elaborated in detail the experimental design and demographics of participants:

6.

Are the implications of this work specific to fear generalization, or relevant to other generalization/transfer learnings? Authors might want to discuss the generalizability of this work.

As the current model has only been applied to a fear conditioning paradigm with visual simple stimuli, it remains unclear to which extent these findings would generalize to other forms of learning or with different stimulus sets. We have added the following sentence on the generalizability of our work at the end of the *Discussion* session (page 15):

"Another limitation of the present research is that our conclusions are based solely on self-report responses. Future studies could overcome this limitation by incorporating multiple response channels, such as physiological and neuronal responses, to investigate generalization behavior. In addition, While the current model has been only utilized in the context of fear conditioning paradigms with one-dimensional stimuli, it would be beneficial for future research to investigate the generalizability of the modeling conclusions to other learning types, stimuli with higher levels of complexity and multiple dimensions, and samples."

7.

Authors can also discuss other possible cognitive mechanisms that can contribute to generalization but were not studied here. For instance, content processing, which is altered in some psychopathologies, and its related pattern separation and pattern completion processes, could be discussed. Can the reported model be used to better understand how context processing affects generalization patterns?

We totally agree that other cognitive mechanisms may be relevant and interesting to explore in future model development. We have added the following sentences to the *Discussion* session (page 15):

"Future research on developing more sophisticated models of generalization behavior should consider additional fundamental cognitive processes that may directly impact these phenomena. For instance, attentional processes, which

allow for the selection and prioritization of relevant information, have been highlighted as a crucial modulator of learning and generalization (Bedard and Song, 2013; Dayan et al., 2000). Additionally, rule-based generalization based on specific relational differences between stimulus features has been shown to occur alongside distance-based generalization (Dunsmoor and Murphy, 2015; Nosofsky and Zaki, 2002). Recent research has also explored the coexistence of these two types of generalization processes and their interaction in producing generalization behavior (Schlegelmilch et al., 2020; Zaman et al., 2022)."

8.

In the generalization literatures, the novel “similar” stimuli are usually termed “generalization stimuli” and not “test stimuli” as in the current report. Making the terminology more consistent with the literature would be helpful.

We appreciate the Reviewer’s suggestion to use "generalization stimuli" instead of "test stimuli" to align with the terminology more commonly used in the generalization literature. However, as pointed out in a recent review the use of 'generalization stimuli' may lead to confusion regarding the observable (the explanans) and the latent mechanisms (the explanandum) (Zaman, Chalkia, et al., 2021). Therefore, we prefer to use "test stimuli" in this study. We have added a footnote on page 1 to clarify the rationale for our choice of terminology when the term "test stimuli" is first introduced. We hope that this explanation will help readers understand the reasoning behind our choice of terminology.

"To avoid confusion between the explanans and the explanandum, we opted to use the term "test stimuli" rather than the more commonly used "generalization stimuli" (Zaman, Chalkia, et al., 2021). Like other cognitive processes (e.g., attention, memory, perception, etc.), generalization can refer to both the observed cognitive and behavioral phenomena (i.e., explanandum) and the underlying mechanisms that cause them (i.e., the explanans). As a phenomenon, generalization refers to the pattern of responding similarly to a range of physically different stimuli. As an underlying mechanism, generalization refers to the process underlying the observed transfer of learning ."

9.

In the introduction, the first and second limitations seem to be very overlapping. To understand the various possible mechanisms that can drive generalization patterns (e.g., perceptual discrimination; limitation 1), one needs to utilize multi-source data (e.g., self-reported similarity judgement; limitation 2). I’d suggest that these limitations

are combined, or alternatively, the distinction is clarified.

We agree with the Reviewer that the two limitations are overlapping in a way that in order to understand the various possible mechanisms that can drive generalization patterns, such as perceptual discrimination, one has to utilize multi-source data, such as self-reported similarity judgments. To address this issue, we have revised the linking words in the relevant paragraphs to make the distinction between the two limitations clearer and to avoid any potential confusion for readers.

10.

“An abundance of empirical evidence has shown that individual differences in learning and perception may account for the diversity in generalization behavior” — In light of the motivation of this work to study individual differences, I suggest elaborating on the existing literature that studied these individual differences. Further, in the discussion, the authors could explain how/if the conclusions from this work fit this existing literature.

We thank the Reviewer for the suggestion and have elaborated more upon past research both in the *Introduction* session, *Results summary* and *Discussion* section where we position our findings with past research.

References

- Bartlema, A., Lee, M., Wetzels, R., & Vanpaemel, W. (2014). A Bayesian hierarchical mixture approach in individual differences: Case studies in selective attention and representation in category learning. *Journal of Mathematical Psychology*, *59*, 132–150. <https://doi.org/10.1016/j.jmp.2013.12.002>
- Bedard, P., & Song, J.-H. (2013). Attention modulates generalization of visuo-motor adaptation. *Journal of Vision*, *13*(12), 12–12. <https://doi.org/10.1167/13.12.12>
- Bennett, D., Silverstein, S. M., & Niv, Y. (2019). The two cultures of computational psychiatry. *JAMA Psychiatry*, *76*(6), 563. <https://doi.org/10.1001/jamapsychiatry.2019.0231>
- Brady, T. F., & Alvarez, G. A. (2011). Hierarchical encoding in visual working memory: Ensemble statistics bias memory for individual items. *Psychological Science*, *22*(3), 384–392. <https://doi.org/10.1177/0956797610397956>
- Colombo, M., & Seriès, P. (2012). Bayes in the brain—on Bayesian modelling in neuroscience. *The British Journal for the Philosophy of Science*, *63*(3), 697–723. <https://doi.org/10.1093/bjps/axr043>
- Corlett, P. R., Horga, G., Fletcher, P. C., Alderson-Day, B., Schmack, K., & Powers, A. R. (2019). Hallucinations and strong priors. *Trends in Cognitive Sciences*, *23*(2), 114–127. <https://doi.org/10.1016/j.tics.2018.12.001>
- Dayan, P., Kakade, S., & Montague, P. R. (2000). Learning and selective attention. *Nature Neuroscience*, *3*, 1218–1223. <https://doi.org/10.1038/81504>
- Dibbets, P., & Evers, E. A. T. (2017). The influence of state anxiety on fear discrimination and extinction in females. *Frontiers in Psychology*, *08*. <https://doi.org/10.3389/fpsyg.2017.00347>
- Dibbets, P., van den Broek, A., & Evers, E. A. T. (2015). Fear conditioning and extinction in anxiety- and depression-prone persons. *Memory*, *23*(3), 350–364. <https://doi.org/10.1080/09658211.2014.886704>
- Duits, P., Cath, D. C., Lissek, S., Hox, J. J., Hamm, A. O., Engelhard, I. M., van den Hout, M. A., & Baas, J. M. P. (2015). Updated meta-analysis of fear conditioning in anxiety disorders. *Depression and Anxiety*, *32*(4), 239–253. <https://doi.org/10.1002/da.22353>
- Dunsmoor, J. E., & Murphy, G. L. (2015). Categories, concepts, and conditioning: How humans generalize fear. *Trends in Cognitive Sciences*, *19*(2), 73–77. <https://doi.org/10.1016/j.tics.2014.12.003>
- Dymond, S., Dunsmoor, J. E., Vervliet, B., Roche, B., & Hermans, D. (2015). Fear generalization in humans: Systematic review and implications for anxiety disorder research. *Behavior Therapy*, *46*(5), 561–582. <https://doi.org/10.1016/j.beth.2014.10.001>
- Frässle, S., Yao, Y., Schöbi, D., Aponte, E. A., Heinzle, J., & Stephan, K. E. (2018). Generative models for clinical applications in computational

- psychiatry. *WIREs Cognitive Science*, 9(3). <https://doi.org/10.1002/wcs.1460>
- Fraunfelter, L., Gerdes, A., & Alpers, G. (2022). Fear one, fear them all: A systematic review and meta-analysis of fear generalization in pathological anxiety. *Neuroscience & Biobehavioral Reviews*, 139, 104707. <https://doi.org/10.1016/j.neubiorev.2022.104707>
- Gibson, E. J. (1953). Improvement in perceptual judgments as a function of controlled practice or training. *Psychological Bulletin*, 50(6), 401–431. <https://doi.org/10.1037/h0055517>
- Gross, S. (2020). Probabilistic representations in perception: Are there any, and what would they be? *Mind & Language*, 35(3), 377–389. <https://doi.org/10.1111/mila.12280>
- Guttman, N., & Kalish, H. I. (1956). Discriminability and stimulus generalization. *Journal of Experimental Psychology*, 51(1), 79–88. <https://doi.org/10.1037/h0046219>
- Hanson, H. M. (1959). Effects of discrimination training on stimulus generalization. *Journal of Experimental Psychology*, 58(5), 321–334. <https://doi.org/10.1037/h0042606>
- Hearst, E. (1968). Discrimination learning as the summation of excitation and inhibition. *Science*, 162(3859), 1303–1306. <https://doi.org/10.1126/science.162.3859.1303>
- Hermann, C., Ziegler, S., Birbaumer, N., & Flor, H. (2002). Psychophysiological and subjective indicators of aversive pavlovian conditioning in generalized social phobia. *Biological Psychiatry*, 52(4), 328–337. [https://doi.org/10.1016/S0006-3223\(02\)01385-9](https://doi.org/10.1016/S0006-3223(02)01385-9)
- Hull, C. L. (1947). The problem of primary stimulus generalization. *Psychological Review*, 54(3), 120–134. <https://doi.org/10.1037/h0061159>
- Insel, T., Cuthbert, B., Garvey, M., Heinssen, R., Pine, D. S., Quinn, K., Sanislow, C., & Wang, P. (2010). Research domain criteria (RDoC): Toward a new classification framework for research on mental disorders. *American Journal of Psychiatry*, 167(7), 748–751. <https://doi.org/10.1176/appi.ajp.2010.09091379>
- Jenkins, H. M., & Harrison, R. H. (1962). Generalization gradients of inhibition following auditory discrimination learning. *Journal of the Experimental Analysis of Behavior*, 5(4), 435–441. <https://doi.org/10.1901/jeab.1962.5-435>
- Jovanovic, T., Norrholm, S. D., Blanding, N. Q., Davis, M., Duncan, E., Bradley, B., & Ressler, K. J. (2010). Impaired fear inhibition is a biomarker of PTSD but not depression. *Depression and Anxiety*, 27(3), 244–251. <https://doi.org/10.1002/da.20663>
- Kaczurkin, A. N., & Lissek, S. (2013). Generalization of conditioned fear and obsessive-compulsive traits. *Journal of Psychology & Psychotherapy*, 7, 3. <https://www.ncbi.nlm.nih.gov/pmc/articles/PMC3932061/>
- Kalish, H. I. (1958). The relationship between discriminability and generalization: A re-evaluation. *Journal of Experimental Psychology*, 55(6), 637–644. <https://doi.org/10.1037/h0048049>

- Knill, D. C., & Alexandre, P. (2004). The Bayesian brain: The role of uncertainty in neural coding and computation. *Trends in Neurosciences*, *27*(12), 712–719. <https://doi.org/10.1016/j.tins.2004.10.007>
- Laufer, O., Israeli, D., & Paz, R. (2016). Behavioral and neural mechanisms of overgeneralization in anxiety. *Current Biology*, *26*(6), 713–722. <https://doi.org/10.1016/j.cub.2016.01.023>
- Lee, M. D. (2011). How cognitive modeling can benefit from hierarchical Bayesian models. *Journal of Mathematical Psychology*, *55*(1), 1–7. <https://doi.org/10.1016/j.jmp.2010.08.013>
- Lee, M. D. (2018). Bayesian methods in cognitive modeling. In J. T. Wixted (Ed.), *Stevens' Handbook of Experimental Psychology and Cognitive Neuroscience* (3rd ed., pp. 1–48). John Wiley & Sons, Inc. <https://doi.org/10.1002/9781119170174.epcn502>
- Lee, M. D., & Vanpaemel, W. (2008). Exemplars, prototypes, similarities, and rules in category representation: An example of hierarchical Bayesian analysis. *Cognitive Science*, *32*(8), 1403–1424. <https://doi.org/10.1080/03640210802073697>
- Lee, M. D., & Wagenmakers, E.-J. (2005). Bayesian statistical inference in psychology: Comment on trafimow (2003). *Psychological Review*, *112*(3), 662–668. <https://doi.org/10.1037/0033-295X.112.3.662>
- Lissek, S., Bradford, D. E., Alvarez, R. P., Burton, P., Espensen-Sturges, T., Reynolds, R. C., & Grillon, C. (2014). Neural substrates of classically conditioned fear-generalization in humans: A parametric fMRI study. *Social Cognitive and Affective Neuroscience*, *9*(8), 1134–1142. <https://doi.org/10.1093/scan/nst096>
- Lissek, S., Powers, A. S., McClure, E. B., Phelps, E. A., Woldehawariat, G., Grillon, C., & Pine, D. S. (2005). Classical fear conditioning in the anxiety disorders: A meta-analysis. *Behaviour Research and Therapy*, *43*(11), 1391–1424. <https://doi.org/10.1016/j.brat.2004.10.007>
- Luck, S. J., & Vogel, E. K. (2013). Visual working memory capacity: From psychophysics and neurobiology to individual differences. *Trends in Cognitive Sciences*, *17*(8), 391–400. <https://doi.org/10.1016/j.tics.2013.06.006>
- Maia, T. V., & Frank, M. J. (2011). From reinforcement learning models to psychiatric and neurological disorders. *Nature Neuroscience*, *14*(2), 154–162. <https://doi.org/10.1038/nn.2723>
- Mkrtchian, A., Aylward, J., Dayan, P., Roiser, J. P., & Robinson, O. J. (2017). Modeling avoidance in mood and anxiety disorders using reinforcement learning. *Biological Psychiatry*, *82*(7), 532–539. <https://doi.org/10.1016/j.biopsych.2017.01.017>
- Nosofsky, R. M., & Zaki, S. R. (2002). Exemplar and prototype models revisited: Response strategies, selective attention, and stimulus generalization. *Journal of Experimental Psychology: Learning, Memory, and Cognition*, *28*(5), 924–940. <https://doi.org/10.1037/0278-7393.28.5.924>

- Okada, K., & Lee, M. D. (2016). A Bayesian approach to modeling group and individual differences in multidimensional scaling. *Journal of Mathematical Psychology*, *70*, 35–44. <https://doi.org/10.1016/j.jmp.2015.12.005>
- Petzschner, F. H., Glasauer, S., & Stephan, K. E. (2015). A Bayesian perspective on magnitude estimation. *Trends in Cognitive Sciences*, *19*(5), 285–293. <https://doi.org/10.1016/j.tics.2015.03.002>
- Plaisted, K. C. (2001). Reduced generalization in autism: An alternative to weak central coherence. In J. A. Burack, T. Charman, N. Yirmiya, & P. R. Zelazo (Eds.), *The Development of Autism* (1st ed., pp. 149–169). Routledge. <https://doi.org/10.4324/9781410600196-15>
- Powers, A. R., Mathys, C., & Corlett, P. R. (2017). Pavlovian conditioning–induced hallucinations result from overweighting of perceptual priors. *Science*, *357*(6351), 596–600. <https://doi.org/10.1126/science.aan3458>
- Press, C., Kok, P., & Yon, D. (2020). The perceptual prediction paradox. *Trends in Cognitive Sciences*, *24*(1), 13–24. <https://doi.org/10.1016/j.tics.2019.11.003>
- Press, C., & Yon, D. (2019). Perceptual prediction: Rapidly making sense of a noisy world. *Current Biology*, *29*(15), 751–753. <https://doi.org/10.1016/j.cub.2019.06.054>
- Raviv, L., Lupyan, G., & Green, S. C. (2022). How variability shapes learning and generalization. *Trends in Cognitive Sciences*, *26*(6), 462–483. <https://doi.org/10.1016/j.tics.2022.03.007>
- Razran, G. (1955). Conditioning and perception. *Psychological Review*, *62*(2), 83–95. <https://doi.org/10.1037/h0046875>
- Rouhani, N., Wimmer, G. E., Schneier, F. R., Fyer, A. J., Shohamy, D., & Simpson, H. B. (2019). Impaired generalization of reward but not loss in obsessive–compulsive disorder. *Depression and Anxiety*, *36*(2), 121–129. <https://doi.org/10.1002/da.22857>
- Sagi, D. (2011). Perceptual learning in vision research. *Vision Research*, *51*(13), 1552–1566. <https://doi.org/10.1016/j.visres.2010.10.019>
- Scheibehenne, B., & Pachur, T. (2015). Using Bayesian hierarchical parameter estimation to assess the generalizability of cognitive models of choice. *Psychonomic Bulletin & Review*, *22*(2), 391–407. <https://doi.org/10.3758/s13423-014-0684-4>
- Schlegelmilch, R., Wills, A., & von Helversen, B. (2020). A cognitive category-learning model of rule abstraction, attention learning, and contextual modulation. *Psychological Review*. <https://doi.org/10.1037/rev0000321>
- Schroijen, M., Pappens, M., Schruers, K., Van den Bergh, O., Vervliet, B., & Van Diest, I. (2015). Generalization of fear to respiratory sensations. *Behavior Therapy*, *46*(5), 611–626. <https://doi.org/10.1016/j.beth.2015.05.004>
- Shepard, R. N. (1957). Stimulus and response generalization: A stochastic model relating generalization to distance in psychological space. *Psychometrika*, *22*(4), 325–345. <https://doi.org/10.1007/BF02288967>

- Shepard, R. N. (1987). Toward a universal law of generalization for psychological science. *Science*, *237*(4820), 1317–1323. <https://doi.org/10.1126/science.3629243>
- Stephan, K. E., & Mathys, C. (2014). Computational approaches to psychiatry. *Current Opinion in Neurobiology*, *25*, 85–92. <https://doi.org/10.1016/j.conb.2013.12.007>
- Struyf, D., Zaman, J., Vervliet, B., & Van Diest, I. (2015). Perceptual discrimination in fear generalization: Mechanistic and clinical implications. *Neuroscience & Biobehavioral Reviews*, *59*, 201–207. <https://doi.org/10.1016/j.neubiorev.2015.11.004>
- Thomas, D. R., & Switalski, R. W. (1966). Comparison of stimulus generalization following variable-ratio and variable-interval training. *Journal of Experimental Psychology*, *71*(2), 236–240. <https://doi.org/10.1037/h0022880>
- Vanpaemel, W., & Lee, M. D. (2012). Using priors to formalize theory: Optimal attention and the generalized context model. *Psychonomic Bulletin & Review*, *19*(6), 1047–1056. <https://doi.org/10.3758/s13423-012-0300-4>
- Wang, X.-J., & Krystal, J. H. (2014). Computational psychiatry. *Neuron*, *84*(3), 638–654. <https://doi.org/10.1016/j.neuron.2014.10.018>
- Watanabe, T., Náñez, J. E., & Sasaki, Y. (2001). Perceptual learning without perception. *Nature*, *413*(6858), 844–848. <https://doi.org/10.1038/35101601>
- Weiss, Y., Simoncelli, E. P., & Adelson, E. H. (2002). Motion illusions as optimal percepts. *Nature Neuroscience*, *5*(6), 598–604. <https://doi.org/10.1038/nn0602-858>
- Wiecki, T. V., Poland, J., & Frank, M. J. (2015). Model-based cognitive neuroscience approaches to computational psychiatry: Clustering and classification. *Clinical Psychological Science*, *3*(3), 378–399. <https://doi.org/10.1177/2167702614565359>
- Winsberg, S., & De Soete, G. (1993). A latent class approach to fitting the weighted Euclidean model, clascal. *Psychometrika*, *58*(2), 315–330. <https://doi.org/10.1007/BF02294578>
- Zaman, J., Vanpaemel, W., Aelbrecht, C., Tuerlinckx, F., & Vlaeyen, J. (2017). Biased pain reports through vicarious information: A computational approach to investigate the role of uncertainty. *Cognition*, *169*, 54–60. <https://doi.org/10.1016/j.cognition.2017.07.009>
- Zaman, J., Ceulemans, E., Hermans, D., & Beckers, T. (2019). Direct and indirect effects of perception on generalization gradients. *Behaviour Research and Therapy*, *114*, 44–50. <https://doi.org/10.1016/j.brat.2019.01.006>
- Zaman, J., Chalkia, A., Zenses, A.-K., Bilgin, A. S., Beckers, T., Vervliet, B., & Boddez, Y. (2021). Perceptual variability: Implications for learning and generalization. *Psychonomic Bulletin & Review*, *28*(1), 1–19. <https://doi.org/10.3758/s13423-020-01780-1>
- Zaman, J., Struyf, D., Ceulemans, E., Vervliet, B., & Beckers, T. (2021). Perceptual errors are related to shifts in generalization of conditioned re-

- sponding. *Psychological Research*, 85(4), 1801–1813. <https://doi.org/10.1007/s00426-020-01345-w>
- Zaman, J., Wiech, K., & Vlaeyen, J. W. (2020). Perceptual Decision Parameters and Their Relation to Self-Reported Pain: A Drift Diffusion Account. *The Journal of Pain*, 21(3-4), 324–333. <https://doi.org/10.1016/j.jpain.2019.06.009>
- Zaman, J., Yu, K., & Lee, J. C. (2022). Individual differences in stimulus identification, rule induction, and generalization of learning. *Journal of Experimental Psychology: Learning, Memory, and Cognition*. <https://doi.org/10.1037/xlm0001153>
- Zenses, A.-K., Lee, J. C., Plaisance, V., & Zaman, J. (2021). Differences in perceptual memory determine generalization patterns. *Behaviour Research and Therapy*, 136, 103777. <https://doi.org/10.1016/j.brat.2020.103777>
- Zhang, L., Lu, X., Bi, Y., & Hu, L. (2019). Pavlov's pain: The effect of classical conditioning on pain perception and its clinical implications. *Current Pain and Headache Reports*, 23(3), 19. <https://doi.org/10.1007/s11916-019-0766-0>

24th May 23

Dear Mr Yu,

Your manuscript titled "Multiple pathways to widespread fears: Disentangling idiosyncratic fear generalization mechanisms using computational modeling" has now been seen by our reviewers, whose comments appear below. In light of their advice I am delighted to say that we are happy, in principle, to publish a suitably revised version in *Communications Psychology* under the open access CC BY license (Creative Commons Attribution v4.0 International License).

We therefore invite you to revise your paper one last time to address a list of editorial requests. At the same time we ask that you edit your manuscript to comply with our format requirements and to maximise the accessibility and therefore the impact of your work.

EDITORIAL REQUESTS:

SUBMISSION INFORMATION:

OPEN ACCESS:

Communications Psychology is a fully open access journal. Articles are made freely accessible on publication under a [CC BY license](http://creativecommons.org/licenses/by/4.0) (Creative Commons Attribution 4.0 International License). This license allows maximum dissemination and re-use of open access materials and is preferred by many research funding bodies.

For further information about article processing charges, open access funding, and advice and support from Nature Research, please visit <https://www.nature.com/commspsychol/article-processing-charges>

At acceptance, you will be provided with instructions for completing this CC BY license on behalf of all authors. This grants us the necessary permissions to publish your paper. Additionally, you will be asked to declare that all required third party permissions have been obtained, and to provide billing information in order to pay the article-processing charge (APC).

* **DATA AVAILABILITY:**

[link redacted]

Best regards,

Marike

Marike Schiffer, PhD
Chief Editor
Communications Psychology

REVIEWERS' COMMENTS:

Reviewer #1 (Remarks to the Author):

I am pleased with the revision and have no further comments.

Reviewer #2 (Remarks to the Author):

The authors did a very good job at revising their manuscript. The changes made have significantly addressed some of the concerns I had raised in my previous review. The paper now provides a clearer and more comprehensive account of the model and its implications.

Reviewer #3 (Remarks to the Author):

I appreciate the significant effort that the authors have made during the revision process. While most of my comments have been addressed, there are a couple of remaining points:

1- To follow-up on my previous comment #7: As the authors mention in a few places in the manuscript, learning about context (i.e., context processing), is involved in generalization. I think that the new text in the “Future directions” section should also include a discussion on context processing as another possible cognitive mechanism, together with the related pattern separation and pattern completion processes.

2- Pg. 14: “Computational methods have gained prominence in exploring these mechanisms, particularly through parameterizing specific psychological processes such as learning 70, 71.”: In light of the focus on individual differences and personality traits in healthy individuals in this manuscript, I suggest to add a reference to demonstrate that computational modeling can also be used to study these, e.g., Sheynin et al. 2015 Behav Brain Res. (PMID: 25639540).

Minor:

- Pg. 2: “Conditioned responses to **test stimuli** were strong...”: replace with “TS’s”.

- Pg. 5: “... but could emerge from **a** several different latent processes...”: remove “a”.